# Combining ability and heterosis analysis for mineral content in the leafy vegetable *Gynandropsis gynandra* (L.) Briq.

Aristide Carlos Houdegbe[1,2*], Enoch G. Achigan-Dako[1], E. O. Dêêdi Sogbohossou[1], Alfred O. Odindo[2], M. Eric Schranz[3], Julia Sibiya[2,4]

1 Genetics, Biotechnology and Seed Science Unit (GBioS), Laboratory of Crop Production, Physiology and Plant Breeding, Faculty of Agricultural Sciences, University of Abomey-Calavi, Abomey-Calavi, Republic of Benin, 2 School of Agriculture and Science, University of KwaZulu-Natal, Pietermaritzburg, Republic of South Africa, 3 Biosystematics Group, Wageningen University, Wageningen, The Netherlands, 4 Sustainable Agrifood Systems Program (SAS), International Maize and Wheat Improvement Center (CIMMYT), Nairobi, Kenya

* houdariscarl@gmail.com

## Abstract

Spider plant (*Gynandropsis gynandra*) is a leafy vegetable rich in micronutrients, including minerals, vitamins, and secondary metabolites, making it a valuable opportunity crop for combating hidden hunger and promoting human health. However, knowledge of the inheritance of mineral content is limited, which hinders the development of improved cultivars for wider cultivation. To address this, 118 $F_1$ experimental hybrids involving 26 parental lines were generated from a North Carolina mating design II. The $F_1$s and their parents were evaluated across two years (2019 and 2020) for gene action, combining ability effects and heterosis of leaf mineral (zinc, copper, manganese, calcium, magnesium, sodium, phosphorus, and potassium) content. Significant differences (p < 0.001) were observed among and between hybrids and parents for iron, zinc, copper, manganese, calcium, magnesium, sodium, phosphorus, and potassium contents. The genotype × year interaction was also significant, with variance greater than the genotypic variance. Significant general and specific combining ability effects, together with variance components analysis, revealed that both additive and nonadditive gene action controlled mineral content, with a predominance of nonadditive gene action. Mid- and best-parent heterosis ranged from -80.4% to 389.5% for mineral content. Parents with good general combining ability were identified, as well as crosses with high specific combining ability and heterosis. There were significant and moderate to strong correlations between mean hybrid performance, specific combining ability effects, and heterosis levels, and low to moderate correlations between general combining ability and the performance of the mean parents. We conclude that hybridization in *G. gynandra* contributes to improving the mineral content. *G. gynandra* can be used as a model crop to study the genetic mechanism underlying heterosis in leafy vegetables.

**Data availability statement:** All relevant data are within the manuscript and its Supporting Information files.

**Funding:** This work was supported by the "Intra-Africa Academic Mobility Scheme" under project grant number 2016-2988 on "Enhancing training and research mobility for novel crops breeding in Africa (MoBreed)" funded by the Education, Audiovisual and Culture Executive Agency (EACEA) of the European Commission through a PhD scholarship awarded to Aristide Carlos Houdegbe. The scholarship was for academic training and research mobility and a research grant to complete a PhD degree at the University of KwaZulu-Natal (South Africa). The funders had no role in study design, data collection and analysis, decision to publish, or preparation of the manuscript.

**Competing interests:** The authors have declared that no competing interests exist.

## Introduction

The nutritive value of vegetables, particularly orphan, underutilized, or opportunity vegetables, is high, and they are an essential source of micronutrients. The increasing interest in leafy vegetables is particularly due to their distinct richness in minerals, vitamins, phytochemicals, and antioxidants [1–3], as well as their good adaptation to local conditions, representing a reliable resource for adapting to a changing climate. The nutritive richness makes leafy vegetables valuable crops to diversify the world's food portfolio, which, to date, is limited to a few crops [4]. Ongoing actions to promote opportunity crops include the development of genomic tools to speed up new cultivar development to meet consumers' preferences [5], as well as raising population awareness of their potential value and the Vision for Adapted Crops and Soils (VACS) [6]. In the case of leafy vegetables, the main breeding objective is to develop high leaf-yielding cultivars [7], to optimize resource use and availability. This objective might conflict or align with the primary potential of leafy vegetables, which is their high micronutrient content. Marles [8] provided evidence of a decline in mineral content in crops, including vegetables. For example, a drop of 10% to 52% of minerals content in fruits and vegetables was observed in the United Kingdom from 1940 to 2019 [9]. The decrease in mineral content is referred to as the "dilution effect", which is an increase in yield in modern cultivars without a subsequent increase in mineral content or higher mineral nutrient concentrations in old cultivars than in improved cultivars. The decline was observed to be more pronounced in vegetables than in other commodity groups [10]. Therefore, assessing the nutritional value of the breeding material throughout the breeding process is essential for the conservation of the primary benefits of leafy vegetables while increasing their productivity.

*Gynandropsis gynandra* (L.) Briq. (syn. *Cleome gynandra* L.), the commonly known spider plant is interesting because of its high content of vitamins, minerals, and secondary metabolites [2,11–17]. The major vitamins reported in the species include vitamins C, A, and E, and minerals include microelements (iron, zinc, copper, manganese) and macrominerals (calcium, potassium, magnesium, phosphorus, and sodium) [11,12,17]. Minerals and vitamins, referred to as micronutrients, are essential in very small quantities and play a crucial role in maintaining health. Although all micronutrients are necessary for human health and bodily function, potassium, calcium, iron, zinc, and magnesium are of global health concern as their deficiencies affect billions of people worldwide [18]. The consumption of spider plant leaves could provide 10–100% of the recommended dietary allowance for human body needs, depending on the age group and the mineral [17]. The species also contains several secondary metabolites, including flavonoids, terpenoids, tannins, glucosinolates, and various phenolic compounds, which are essential for human health [2,12,13,19,20]. All these nutritional and health-promoting compounds exhibit a wide range of variation in the species, providing a strong basis for crop improvement [11,12,16,17,19].

More importantly, *G. gynandra* leaves have 4.7- and 3.2-fold of vitamin C, 4- and 2-fold iron, 1.2- and 2.1-fold zinc, 2.7- and 10.4-fold calcium, 3.3- and 5.5-fold phosphorus, 1.4- and 1.8-fold potassium, 2.59- and 1.37-fold total phenolics, and 5.70- and 2.46-fold flavonoids concentrations higher than two world leading commercial

and consumed vegetables namely cabbage (*Brassica oleracea* var. *capitata* cv. Drumhead) and Swiss chard (*Beta vulgaris* L. cv. Fordhook Giant), respectively [2]. Breeding efforts in *G. gynandra* will result in significant impacts on healthier and balanced diets for local communities in Africa and Asia where the species is mostly consumed [7,21] and high micronutrient deficiencies occurs. Moreover, *G. gynandra* is a climate resilient crop as it is a $C_4$ plant with the ability to withstand various harsh conditions [21]; the species was used as a model crop to understand $C_4$ traits, evolution and gene expression in plants [22–24]. The production and the exploitation of the nutritional potential of *G. gynandra* like many other opportunity crops are limited due to lack of improved varieties resulting from the lack of sustainable breeding program.

Breeding strategies and type of cultivars for a given crop species are guided by the species reproductive biology and mating systems. Both self- and cross-compatibility occurs in *G. gynandra* with outcrossing being predominant [25–27], offering the possibility for developing both hybrids and inbred cultivars. Because of the predominance of outcrossing, the species might exhibit heterosis and the choice for developing hybrids cultivars sounds as good strategy to exploit the potential heterosis that needs to be properly documented. Heterosis or hybrid vigour refers to the outperformance of the first generation of progenies compared to their parents [28–31] and has been highly researched by breeders for open-pollinated crops but also for self-pollinated crops for higher productivity. Four steps are crucial in the development of a hybrid cultivar: (i) the establishment of populations for selection; (ii) inbred lines development; (iii) inbred lines' evaluation for combining ability; and (iv) the production of hybrid seed [32]. Following the development of the inbred/advanced lines in *G. gynandra* [17,33], their testing for combining ability is the next step towards hybrids creation. Combining ability refers to the ability of a line to combine with another one during hybridization so that desirable genes or traits are transferred to their progenies and encompasses two types, namely the general combining ability (GCA) and the specific combining ability (SCA) [34]. According to Sprague and Tatum [34], a GCA of a line is the average performance of this line in a set of its hybrid combinations and the SCA is the deviation of a specific cross performance from the sum of the average performance or GCA of its parental lines. Combining ability informs on the nature and magnitude of gene action controlling the considered trait. GCA are associated with additive genes effects, while the SCA effects are attributed to nonadditive gene action, including dominance and epistasis gene effects [34]. The GCA helps in the selection of good parents for breeding and the SCA is useful for the selection of the best hybrid combination to better exploit heterosis. The assessment of the combining ability of a set of lines and the inheritance patterns for a target species is done using mating designs such as North Carolina Design II, diallel and line by tester, among others [35–37].

*G. gynandra* belongs to Cleomaceae family, a sister family of Brassicaceae and could gain more from breeding strategies implemented in *Brassica* crops. Most developed varieties for *Brassica* crops nowadays are hybrids and the evaluation of combining ability, heterosis level and gene action has been intensively done for several parents in *Brassica oleracea* var. *capitata* [38], *Brassica rapa* L. [39], and *Brassica oleracea* var. *botrytis* L. [40,41] for nutritional traits such as minerals, antioxidant pigments, and vitamins. Unfortunately, to the best of our knowledge, the heterosis level and the combining ability potential for mineral content in *G. gynandra* are yet to be investigated to inform breeding program of good parents, heterotic crosses and breeding methods for improved cultivars development.

In this study, we generated knowledge on the genetic mechanism controlling mineral content in *G. gynandra* with a good perspective for hybrid cultivar development. The present study aimed to investigate the mineral profile of experimental hybrids of spider plant to select high-nutrient hybrids for cultivar development. Specifically, the study (i) compared the mineral content of 118 experimental $F_1$ *G. gynandra* hybrids and their parental lines; (ii) quantified heterosis magnitude for leaf mineral content in *G. gynandra* to select the most suitable breeding method; (iii) evaluated the combining ability effects for leaf mineral content of the parental lines of *G. gynandra* to identify good parents and best crosses; and (iv) estimated the extent of association between mean performance, heterosis and combining ability for mineral content in *G. gynandra* to guide prediction of hybrid performance and multiple traits selection. The following research questions guided this study: (i) How does the leaf mineral content in *G. gynandra* pass from the parents to the progenies? (ii) Does heterosis exist for leaf mineral content in *G. gynandra*, and to what extent? (iii) What could be the breeding strategies for the

development of nutrient-dense cultivars of spider plant? We therefore hypothesized that: (i) the $F_1$ hybrids outperformed their parental lines for leaf mineral content; (ii) the inheritance of leaf mineral content in spider plant are controlled by both additive and non-additive gene action; (iii) there are lines and hybrids with good combining ability effects for leaf mineral content; (iv) estimates of combining ability are good predictors for mean hybrids performance and heterosis level for leaf mineral content in *G. gynandra*; (v) the leaf mineral elements content in *G. gynandra* are correlated; and (vi) the correlations between leaf mineral elements content are stable from parents to hybrids.

## Materials and methods

### Plant material

Twenty-six advanced lines, derived from 26 accessions originating from various Asian and African countries (Table 1) and self-pollinated, were used in this study. The lines were separated into two groups; 12 and 14 lines used as females and males, respectively, based on the male/staminate flowers/pollen and female/hermaphrodite flowers production ability (Table 1) as most individual plants in the species were reported to be andromonoecious with differential flowers

**Table 1. List of 26 advanced lines of *G. gynandra* used as parents to generate the 118 hybrids used in this study and their origin.**

| Lines | Genebank of the original accession | Country of origin | Continent | Parent |
|-------|-----------------------------------|-------------------|-----------|--------|
| P01 | WorldVeg | Malaysia | Asia | Female |
| P02 | WorldVeg | Thailand | Asia | Female |
| P03 | WorldVeg | Malaysia | Asia | Female |
| P04 | WorldVeg | Thailand | Asia | Female |
| P05 | WorldVeg | Malaysia | Asia | Female |
| P06 | WorldVeg | Lao People's Democratic Republic | Asia | Female |
| P07 | WorldVeg | Lao People's Democratic Republic | Asia | Female |
| P08 | WorldVeg | Malawi | Africa | Female |
| P09 | KENRIK | Kenya | Africa | Female |
| P10 | GBioS | Benin | Africa | Female |
| P11 | GBioS | Benin | Africa | Female |
| P12 | GBioS | Benin | Africa | Female |
| P13 | WorldVeg | Lao People's Democratic Republic | Asia | Male |
| P14 | WorldVeg | Lao People's Democratic Republic | Asia | Male |
| P15 | KENRIK | Kenya | Africa | Male |
| P16 | KENRIK | Kenya | Africa | Male |
| P17 | KENRIK | Kenya | Africa | Male |
| P18 | WorldVeg | Kenya | Africa | Male |
| P19 | WorldVeg | Zambia | Africa | Male |
| P20 | WorldVeg | Uganda | Africa | Male |
| P21 | WorldVeg | Uganda | Africa | Male |
| P22 | WorldVeg | Uganda | Africa | Male |
| P23 | WorldVeg | Kenya | Africa | Male |
| P24 | WorldVeg | Zambia | Africa | Male |
| P25 | GBioS | Togo | Africa | Male |
| P26 | GBioS | Togo | Africa | Male |

GBioS: Genetics, Biotechnology and Seed Science Unit, University of Abomey-Calavi; WorldVeg: World Vegetable Center; KENRIK: Kenya Resource Center for Indigenous Knowledge.

productivity [25,26]. Males were crossed with females in a North Carolina design II during two summer seasons (season 1, from October 2018 to February 2019 and season 2, from October 2019 to March 2020) in a greenhouse at the Controlled Environment Facility (29°46′ S, 30°58′ E) of the University of KwaZulu-Natal, Pietermaritzburg Campus, South Africa. Across the two crossing seasons, a total of 118 successful unique single crosses with sufficient seeds were generated for evaluation (S1 Table). In addition, each line was self-pollinated during each crossing season. Crosses were performed following the protocol developed by Zohoungbogbo et al. [42].

## Experimental design and growth conditions

The 26 parental lines and 118 $F_1$ hybrids were grown in 2019 (March to June) and 2020 (September to December) using an alpha design (10 incomplete blocks with 12 entries per incomplete block for hybrids and seven incomplete blocks with four entries per incomplete block for parents) with two replications in a greenhouse at the Controlled Environment Facility of the University of KwaZulu-Natal, Pietermaritzburg Campus, South Africa. The parents were blocked separately from the hybrids. Seeds of all genotypes were pretreated by heating at 40 °C for three days to improve germination before sowing in cell trays filled with growing media. Cell trays were established in the greenhouse, and germination began three days after planting. Seedlings were grown for four weeks, after which they were transplanted.

In 2019, seedlings were transplanted into a single-row plot, 1 m in length, with a spacing of 20 cm between and within rows on raised beds measuring 1 m in width and 1 m in height. The soil had a clay content of 38.5%, a bulk density of 1 g/cm³, and a pH (KCl) of 5.6. The soil contained 78 mg kg$^{-1}$ of phosphorus, 133 mg kg$^{-1}$ of potassium, 2576 mg kg$^{-1}$ of calcium, 384 mg kg$^{-1}$ of magnesium, 22 mg kg$^{-1}$ of zinc, 11 mg kg$^{-1}$ of manganese, 5 mg kg$^{-1}$ of copper, 1900 mg kg$^{-1}$ of nitrogen, and 2% of organic carbon. In 2020, seedlings were transplanted into 10-liter pots with three plants per pot. Pots were filled with composted pine-bark growing media. The growing media was characterized by 31% of carbon, 11 g kg$^{-1}$ of nitrogen, 13 g kg$^{-1}$ of calcium, 3300 mg kg$^{-1}$ of magnesium, 2500 mg kg$^{-1}$ of potassium, 3400 mg kg$^{-1}$ of phosphorus, 470 mg kg$^{-1}$ of sodium, 181 mg kg$^{-1}$ of zinc, 42 mg kg$^{-1}$ of copper, 1034 mg kg$^{-1}$ of manganese and 7374 mg kg$^{-1}$ of iron on dry matter basis.

In both years, weeds were controlled manually while plants were watered daily using an automated drip irrigation with 1 litre per pot. Before transplanting, basal fertilizer composed of N:P: K (2:3:2) at a dose of 150 kg ha$^{-1}$ was applied, and limestone ammonium nitrate (28% N) was used as topdressing two weeks after transplanting at a dose of 100 kg ha$^{-1}$. The greenhouse structure was made of corrugated polycarbonate material, which reduced the outside solar irradiance by 50% on average. Therefore, the maximum of solar radiation was 356 and 567 W m$^{-2}$ in 2019 and 2020, respectively. In 2019, the day temperature varied between 20 and 35°C and the night temperature ranged from 15 to 25°C (S1 Fig). At the same time, the relative humidity was between 75 and 87% at night and 45 and 70% during day hours. In 2020, while the day temperature ranged from 25 and 38°C and the night temperature ranged between 20 and 25°C. The day relative humidity was between 50 and 80% and between 80 and 90% at night (S1 Fig).

## Mineral analysis

A sample of 20 g of tenders and edible fresh leaves per genotype was randomly collected from all the plants in each replicate in paper bags four weeks after transplanting, meaning eight weeks old plants from sowing date. The leaves were picked from plants between 08:00 and 11:00 am. The samples were immediately transported to the laboratory, washed, and oven-dried at 65°C for 72 h.

In 2019, after cooling, the dried leaves were ground into a powder using a mortar and pestle and then sieved through a 1 mm screen sieve. Two independent replicates of 0.5 g each of sieved powder were weighed in porcelain crucibles using an analytical balance (D&T, ES-E200A, max = 200 g, d = 0.1 mg, China). Samples were then ashed in a muffle furnace at 550°C for 2 hours. The obtained ashes were digested using 10 ml of double acid composed of nitric acid (HNO$_3$, 65%, Merck, Germany) and hydrochloric acid (HCl, 32%, Merck, Germany) mixed in a ratio of 1:3 [43].

The resultant mixtures were placed on a hot plate at 250°C for 30 min and later cooled for 1 hour. Digestates were filtered using Whatman paper Grade 1 (Qualitative Filter Paper Standard Grade, circle, 125 mm, Merck, Germany) into a 100 ml volumetric flask and made up to the mark using deionized water. The resultant solutions were analyzed using a fast sequential atomic absorption spectrometer (Varian AA280FS, Varian Inc., Mulgrave, Victoria, Australia) for calcium (Ca), copper (Cu), iron (Fe), zinc (Zn), potassium (K), manganese (Mn), magnesium (Mg) and sodium (Na). Flame atomic absorbance spectroscopy was used for all elements except potassium, for which flame atomic emission spectroscopy was employed. The wavelengths used were 324.8 nm for Cu, 248.3 nm for Fe, 279.5 nm for Mn, 766.5 nm for K, 213.9 nm for Zn, 422.7 nm for Ca, 285.2 nm for Mg, and 589.0 nm for Na. Multielement standard solution IV (23 elements) (1000 mg l$^{-1}$ in HNO$_3$ Suprapur® 6.5%) was purchased from Merck, KGaA, Darmstadt, Germany, and used for calibration. The phosphorus was analyzed in the digested solution according to the 4500-P E ascorbic acid method [44] at 670 nm using an Alpha UV–VIS spectrophotometer (Spectronic Unicam, Berlin, Germany) [45].

In 2020, due to the failure of the atomic absorption spectrometer, the dried plant materials were sent to the Plant Laboratory of the KwaZulu-Natal Department of Agriculture and Rural Development at CEDARA Research station for analysis of leaf elemental composition (Ca, Cu, Fe, K, Mn, Mg, Na, P and Zn). The study was performed based on Hunter [46] with slight modifications [47], using an inductively coupled plasma-optical emission spectrometer (ICP–OES) (Agilent 5800 VDV, Agilent Technologies Australia (M) Pty Ltd. Inc., Mulgrave, Australia). Mineral concentrations were reported in mg kg$^{-1}$ on a dry weight basis (mg kg$^{-1}$ DW) for microelements (Cu, Fe, Mn, and Zn) and in g kg$^{-1}$ on a dry weight basis (g kg$^{-1}$ DW) for macroelements (Ca, K, Mg, P, and Na). Data are presented in S2 Table.

## Statistical analysis

The software R version 4.4.0 [48] was used to perform all statistical analyses. Before proceeding to the analyses, the quality of the data was assessed, mainly for outlier detection with the Bonferroni–Holm test based on studentized residuals at the level of significance of 5%, as recommended by Bernal-Vasquez et al. [49]. Data normality was assessed using the Shapiro–Wilk test. Descriptive statistics (minimum, maximum, mean, coefficient of variation, standard error) were used to characterize the parents and hybrids using the function *describe* from the R package *psych* [50]. The significance of differences between the overall means of parents and hybrids was tested using the *t-test* or *the Wilcoxon test* when necessary. For each mineral content, estimates of variance components were computed per year and across years. Per year, each hybrid and parental dataset was analyzed separately for overall genotypic variance components and adjusted means (best linear unbiased predictors - BLUPs) using the following statistical model 1:

$$y_{ik} = \mu + R_k + G_i + \varepsilon_{ik} \tag{1}$$

where $y_{ik}$ is the phenotypic observation of the $i^{th}$ genotype (hybrid or parental line) in the $k^{th}$ replicate, $\mu$ is the overall mean, $R_k$ is the random effect of the $k^{th}$ replicate, $G_i$ is the random effect of the $i^{th}$ genotype, and $\varepsilon_{ik}$ is the random residual.

Standard broad-sense heritability was calculated according to Hallauer et al. [37] as follows:

$$H^2 = \sigma_G^2 / (\sigma_G^2 + \sigma_e^2 / r) \tag{2}$$

where $\sigma_G^2$ was the total genotypic variance, $\sigma_e^2$ was the residual variance, and $r$ was the number of replications.

Across years, estimates of variance components for overall genotypic effects were computed using the following statistical model:

$$y_{ijk} = \mu + Y_j + R_k (Y_j) + G_i + GY_{ij} + \varepsilon_{ijk} \tag{3}$$

in which $y_{ijk}$ was the phenotypic observation of the $i^{th}$ genotype (hybrid or parental line) in the $k^{th}$ replicate at the $j^{th}$ year, $\mu$ was the overall mean, $Y_j$ was the random effect of the $j^{th}$ year, $R_k(Y_j)$ was the random effect of the $k^{th}$ replicate within the $j^{th}$ year, $G_i$ was the random effect of the $i^{th}$ genotype, $GY_{ij}$ was the random effect of the interaction between the $i^{th}$ genotype and the $j^{th}$ year, and $\varepsilon_{ijk}$ was the random residual. Heterogeneous residual variances were assumed among years. Across years, broad-sense heritability was calculated according to Hallauer et al. [37] using the following formula:

$$H^2 = \sigma_G^2 / (\sigma_G^2 + \sigma_{G \times Y}^2 / n + \sigma_e^2 / nr) \tag{4}$$

where $\sigma_G^2$ was the total genotypic variance, $\sigma_{G \times Y}^2$ was the genotype × year interaction variance, $\sigma_e^2$ was the residual variance, $r$ was the number of replications, and $n$ was the number of years.

Using the hybrids' data only, the following statistical model (5) was used to determine the variance components of the specific combining ability (SCA) and general combining ability (GCA) in each year:

$$Y_{fmk} = \mu + \alpha_f + \beta_m + \gamma_{fm} + r_k + \varepsilon_{fmk} \tag{5}$$

where $y_{fmk}$ was the phenotypic observation of the hybrid between the $f^{th}$ female and $m^{th}$ male in the $k^{th}$ replicate; $\mu$ was the overall mean; $\alpha_f$ was the random GCA effect of the $f^{th}$ female; $\beta_m$ was the random GCA effect of the $m^{th}$ male; $\gamma_{fm}$ was the random SCA effect of the cross between the $f^{th}$ female and the $m^{th}$ male; $r_k$ was the random effect of the $k^{th}$ replicate; and $\varepsilon_{fmk}$ was the random residual.

Across years, the following statistical model (6) was used to determine the variance components of the specific combining ability (SCA) and general combining ability (GCA):

$$Y_{fmjk} = \mu + ry_{kj} + y_j + \alpha_f + \beta_m + \gamma_{fm} + \alpha y_{fj} + \beta y_{mj} + \gamma y_{fmj} + \varepsilon_{fmkj} \tag{6}$$

where $y_{fmjk}$ was the phenotypic observation of the hybrid between the $f^{th}$ female and $m^{th}$ male in the $k^{th}$ replicate at the $j^{th}$ year; $\mu$ was the overall mean; $\alpha_f$ was the random GCA effect of the $f^{th}$ female; $\beta_m$ was the random GCA effect of the $m^{th}$ male; $\gamma_{fm}$ was the random SCA effect of the cross between the $f^{th}$ female and the $m^{th}$ male; $\alpha y_{fj}$ was the random effect of the interaction between the GCA effect of the $f^{th}$ female and the $j^{th}$ year; $\beta y_{mj}$ was the random effect of the interaction between the GCA effect of the $m^{th}$ male and the $j^{th}$ year; $\gamma y_{fmj}$ was the random effect of the interaction between the SCA effect of the cross between the $f^{th}$ female and the $m^{th}$ male and the $j^{th}$ year; $ry_{kj}$ was the random effect of the $k^{th}$ replicate within the $j^{th}$ year; and $\varepsilon_{fmkj}$ was the random residual.

All linear mixed-effects models were fitted using the restricted maximum likelihood (REML) implemented in the "ASReml-R" package version 4.1.0.160 [51]. The likelihood ratio test [52] was used to test the significance of the variance components using the function *lrt* implemented in the ASReml-R package. The additive genetic variance ($\sigma_A^2$), dominance genetic variance ($\sigma_D^2$), total phenotypic variance ($\sigma_P^2$), and narrow-sense ($h^2$) heritability estimates in each year were determined according to Hallauer et al. [37] and Isik et al. [53] as follows:

$$\sigma_A^2 = 2(\sigma_{GCA-M}^2 + \sigma_{GCA-F}^2) \tag{7}$$

$$\sigma_D^2 = 4\sigma_{SCA-F \times M}^2 \tag{8}$$

$$\sigma_P^2 = \sigma_G^2 + \sigma_e^2 / r \tag{9}$$

$$h^2 = \sigma_A^2 / \sigma_P^2 \tag{10}$$

where $\sigma_A^2$ was the additive genetic variance, $\sigma_{GCA-M}^2$ was the male GCA variance, $\sigma_{GCA-F}^2$ was the female GCA variance, $\sigma_D^2$ was the dominance genetic variance, $\sigma_P^2$ was the total phenotypic variance, $\sigma_{SCA-F\times M}^2$ was the SCA variance, $\sigma_e^2$ was the residual variance, and $r$ was the number of replications.

The average degree of dominance was computed as D = $\sqrt{(2\sigma_D^2/\sigma_A^2)}$ [54]. The phenotypic best linear unbiased predictors (BLUPs) associated with the general combining ability effect of each female ($GCA_f$) and male ($GCA_m$) parent and the specific combining ability effect of each cross ($SCA_{fm}$) were derived from model 5 according to Isik et al. [53]. BLUPs associated with the combining ability effects were used due to the incomplete factorial mating design and interest in the family represented by each parental line. In addition, BLUPs were used because of their good predictive accuracy as they have high correlation with the actual values and have been recommended for phenotypic selection in plant breeding [55–57]. The significance of GCA and SCA effects was evaluated using a two-tailed t-test, at the probability levels of 0.05, 0.01, and 0.001 [58]. The importance of dominance and additive gene effects was assessed through the predictability ratio of Baker [59] using the following formula:

$$Predictability\ ratio = (\sigma_{GCA-M}^2 + \sigma_{GCA-F}^2)/(\sigma_{GCA-M}^2 + \sigma_{GCA-F}^2 + \sigma_{SCA-F\times M}^2) \tag{11}$$

where $\sigma_{GCA-M}^2$ was the male GCA variance, $\sigma_{GCA-F}^2$ was the female GCA variance, and $\sigma_{SCA-F\times M}^2$ was the SCA variance.

For heterosis analysis, the adjusted means of each parent and hybrid generated from model 1 were used to estimate the heterosis level. Mid-parent heterosis (MPH) and best-parent heterosis (BPH) were computed for each hybrid as follows:

$$MPH\ (\%) = [(F_1 - MP)/MP] \times 100 \tag{12}$$

$$BPH\ (\%) = [(F_1 - BP)/BP] \times 100 \tag{13}$$

where $F_1$ was the adjusted mean value of the hybrid, MP was the mid-parent adjusted mean value computed as the average adjusted mean values between the two parents of the hybrid, and BP was the adjusted mean value of the best parent.

The genetic advance (GA) for each trait was computed as $GA = i \times H^2 \times \sigma_P$, where $\sigma_P$ was the phenotypic standard deviation, $H^2$ was the broad-sense heritability, and $i$ was the standardized selection differential at the selection intensity of 5% ($i = 2.06$) [60]. Genetic advance over mean (GAM) was further computed as $GAM = (GA/\mu) \times 100$, where $\mu$ is the overall mean of the trait. Spearman correlation coefficients between combining ability, heterosis and mean performance of the parents and hybrids for all characteristics and their level of significance were performed using the function *corr* from the R package *Hmisc* [61].

## Results

### Performance of parents and hybrids

Variations in the leaf contents of iron, zinc, copper, manganese, calcium, potassium, magnesium, phosphorus, and sodium for both parents and hybrids are summarized in Table 2. Trait variability ranged from low to high and depended on the trait (Table 2). For both parents and hybrids, the highest variability was observed for manganese, with a coefficient of variation (CV) of 83% and 94% for parents and hybrids, respectively. In contrast, the lowest coefficient of variation (CV) was observed for potassium (CV = 17%) in parents and for phosphorus in hybrids (21%). For both parents and hybrids, the order of mineral element content in the leaves was potassium > calcium > phosphorus > magnesium > sodium > iron or manganese > zinc > copper.

**Table 2. Descriptive statistics of nine minerals content investigated in 118 hybrids and their 26 parents of *Gynandropsis gynandra* evaluated over two years (2019 and 2020) in a greenhouse.**

| Minerals | Mean | | Minimum | | Maximum | | Standard error | | Coefficient of variation (%) | |
|---|---|---|---|---|---|---|---|---|---|---|
| | Hybrids | Parents | Hybrids | Parents | Hybrids | Parents | Hybrids | Parents | Hybrids | Parents |
| **Macroelements** | | | | | | | | | | |
| Ca (g kg$^{-1}$) | 23.45 | 24.49 | 10.92 | 11.42 | 49.82 | 53.22 | 0.38 | 0.92 | 35.33 | 37.61 |
| Mg (g kg$^{-1}$) | 4.93 | 4.99 | 1.37 | 1.53 | 9.63 | 9.38 | 0.08 | 0.18 | 34.62 | 36.74 |
| K (g kg$^{-1}$) | 34.25 | 25.47 | 16.13 | 12.26 | 63.73 | 34.96 | 0.51 | 0.45 | 32.41 | 17.7 |
| Na (g kg$^{-1}$) | 0.9 | 1.05 | 0.34 | 0.56 | 3.47 | 1.73 | 0.02 | 0.02 | 38.81 | 23.66 |
| P (g kg$^{-1}$) | 9.55 | 7.99 | 3.93 | 4.84 | 15.55 | 13.39 | 0.09 | 0.2 | 20.99 | 25.12 |
| **Microelements** | | | | | | | | | | |
| Fe (mg kg$^{-1}$) | 151.52 | 158.25 | 41.7 | 67.76 | 527.96 | 430.72 | 2.73 | 6.64 | 39.01 | 41.93 |
| Zn (mg kg$^{-1}$) | 73.19 | 56.07 | 27.88 | 28.47 | 224.95 | 104.31 | 1.28 | 1.43 | 37.95 | 25.56 |
| Cu (mg kg$^{-1}$) | 9.66 | 10.63 | 1.73 | 3.03 | 27.04 | 25.1 | 0.15 | 0.35 | 34.61 | 33.16 |
| Mn (mg kg$^{-1}$) | 192.03 | 109.89 | 14.91 | 18.2 | 558.04 | 433.08 | 7.35 | 10.37 | 83 | 94.4 |

Overall, significant differences (p < 0.05) were observed between the mean of parents and hybrids for all microelements (Fe, Zn, Cu, Mn) and phosphorus, potassium, and sodium (Fig 1) in each year and across years, except for iron. Hybrids performed better than their parents for Zn, Mn, P, and K, with average increases of 30.5%, 74.7%, 34.4%, and 19.5%, respectively. In contrast, the mean Na content of the parents was higher (14.3%) than that of the hybrids. While the mean of hybrids was lower than that of parents for Fe in 2019, a higher mean hybrid value was observed in 2020. The inverse was noticed for Cu.

## Variance components and heritability estimates

Combined analysis across years showed that year and genotype by year interaction variances were highly significant for mineral contents in both parents and hybrids (Table 3). Genotypic variances were significant for K and P in parents and for Ca and Mg in hybrids only. Irrespective of the type of population (hybrids or parents), genotype × year interaction variances were higher than the genotypic variance (Table 3). The partitioning of genotypic variance in hybrids showed that general combining ability (GCA) variance for females ($\sigma^2_{GCA-F}$) and males ($\sigma^2_{GCA-M}$) were significantly different from zero for Ca only. Variances due to female general combining ability × year interaction ($\sigma^2_{GCA-F \times Y}$) were significant for all mineral contents except Mn and Mg. In contrast, variances due to male general combining ability × year interaction ($\sigma^2_{GCA-M \times Y}$) were significant for Ca and P. Specific combining ability × year interaction variances ($\sigma^2_{SCA-F \times M \times Y}$) were substantial for all minerals content and greater than any other combining ability variances alone or with year interaction (Table 3).

Given the significant effect of year and its interaction with genotype and combining abilities, estimates of genetic variance components, heritability estimates, degree of dominance, Baker's [59] predictability ratio, and genetic advance were estimated in each year and summarized for macroelements (Ca, Mg, Na, K, P) in Table 4 and microelements (Fe, Mn, Cu, Zn) in Table 5. For all minerals, the genotypic variance was higher than the residual variance in both parents and hybrids each year (Tables 4 and 5). The partitioning of genotypic variance in hybrids showed that general combining ability (GCA) variance for females ($\sigma^2_{GCA-F}$) was significantly different from zero for Ca, P and Zn each year, for Mg, Na, Fe, Cu, and Mn in 2019, and K in 2020 (Tables 4 and 5). The GCA variance for males ($\sigma^2_{GCA-M}$) was significantly different from zero for all macro elements, such as Ca, K, Mg, and P, in both years and for Na in 2019. In contrast, the GCA variance for males ($\sigma^2_{GCA-M}$) was not significant for all microelements (Fe, Cu, Mn, Zn) in 2019 and 2020. While estimates of female GCA variance for Zn were larger than the male ones, male GCA variances were greater than female GCA variances for Mg in both

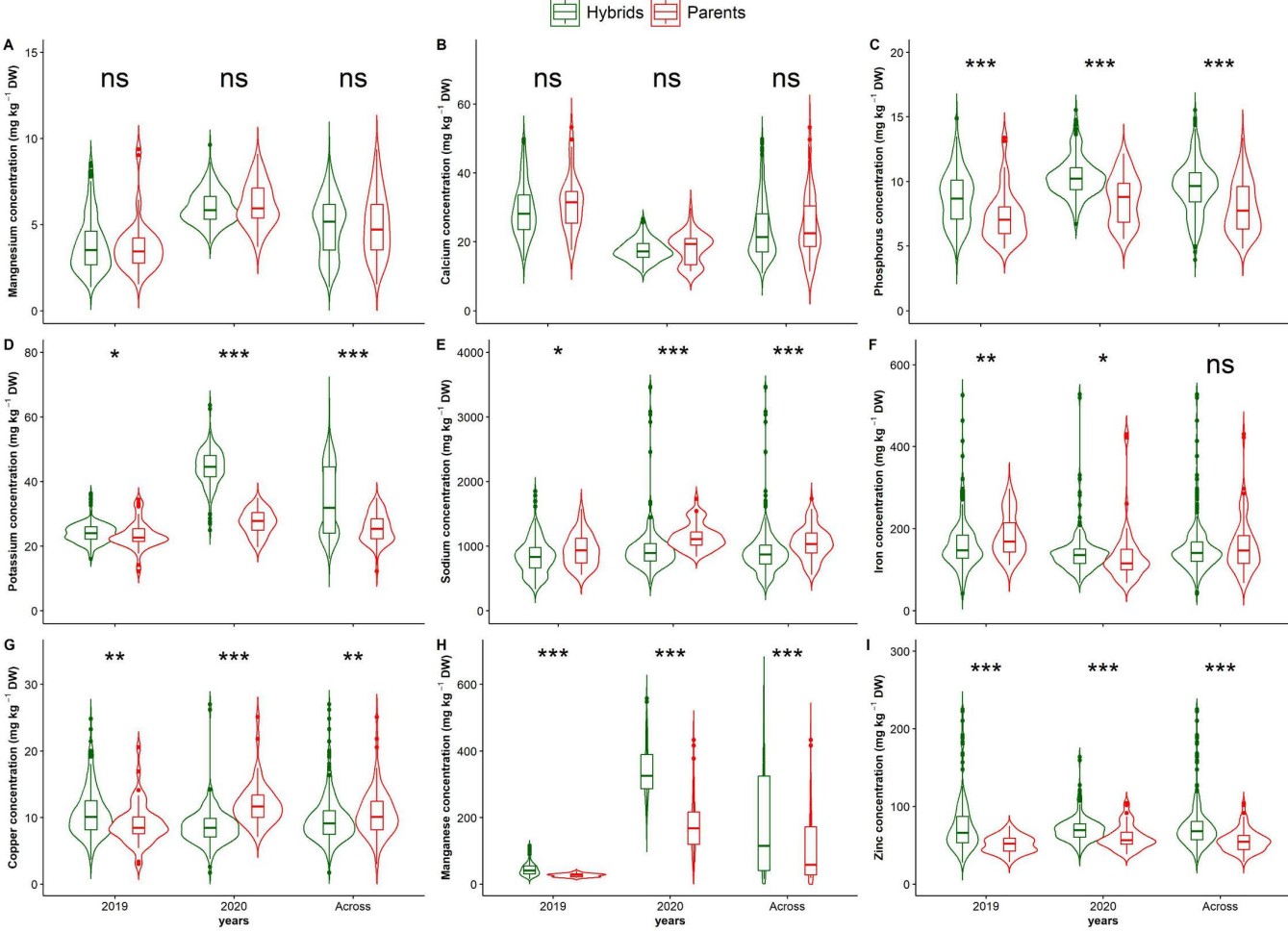

**Fig 1. Comparison of hybrids and parents' performance for leaf mineral content in 118 hybrids and their 26 parents of *G. gynandra* evaluated in 2019 and 2020.** ***, **, * indicate significance at the 0.001, 0.01, and 0.05 probability level, respectively.

years. For all mineral contents, the specific combining ability (SCA) variance ($\sigma^2_{SCA-F \times M}$) was significantly different from zero (p < 0.05) with an estimate greater than the average and sum of females and males GCA variances in both years. Similarly, the additive variance ($\sigma^2_A$) was lower than the dominance variance ($\sigma^2_D$) for all mineral elements. The degree of dominance was greater than unity for all mineral contents, showing their dominant nature. Overall, the predictability ratio was lower than 0.5 for all mineral contents in both years (Tables 4 and 5).

Broad-sense heritability ($H^2$) estimates were moderate to high for all minerals in both hybrids and parents per year, ranging between 0.57 and 0.99 for parents and between 0.81 and 0.99 for hybrids (Tables 4 and 5). In contrast, narrow-sense heritability ($h^2$) estimates were low to moderate. Low $h^2$ values ($\leq$ 0.30) were observed for all microelements each year. Similarly, low $h^2$ values were obtained for macro elements in 2019 and 2020, except Na (0.32), P (0.33), and Ca (0.40) in 2019, for which moderate $h^2$ values were observed. Furthermore, differential $h^2$ values were observed between males and females and varied from year to year (Tables 4 and 5). Genetic gains (> 20% of the current mean of the hybrid population) at 5% selection intensity were significant for all minerals (Tables 4 and 5). The highest genetic gain estimates were 95.90% in 2019 and 82.60% in 2020 for Zn and Na, respectively.

**Table 3. Estimates of variance components for nine mineral elements in 118 hybrids and their 26 parents of *Gynandropsis gynandra* evaluated over two years in a greenhouse.**

| Source variation | Ca | K | Mg | Na | P | Cu | Fe | Mn | Zn |
|---|---|---|---|---|---|---|---|---|---|
| **Parents** | | | | | | | | | |
| $\sigma^2_Y$ | 85.01±122.34*** | 7.91±11.87* | 2.83±4.11*** | 0.02±0.03±ns | 0.69±1.12* | 4.00±6.18* | 1030.48±1649.51* | 12704.95±18081.32*** | 40.60±64.96* |
| $\sigma^2_G$ | 2.65±8.22 ns | 5.97±3.49* | $4.27\ 10^{-07}$±0 ns | $3.08\ 10^{-04}$±$9.49\ 10^{-04}$ ns | 1.3±0.81* | 2.07±1.88 ns | $7.56\ 10^{-04}$±0 ns | 3.13±50.81 ns | 54.63±39.41 ns |
| $\sigma^2_{G\times Y}$ | 35.84±10.86*** | 7.95±2.85*** | 1.87±0.39*** | 0.04±0.01*** | 2.34±0.7*** | 5.07±1.98*** | 2930.04±703.69*** | 13.13±51.15 ns | 122.46±37.51*** |
| $\sigma^2_e$ | 3.09±0.72 | 3.46±0.81 | 0.07±0.01 | $6.54\ 10^{-03}$±$1.56\ 10^{-3}$ | 0.1±0.02 | 3.55±0.82 | 924.12±190.03 | 4156.08±824.46 | 10.71±2.19 |
| $H^2$ | 0.12±0.36 | 0.55±0.19 | $4.45\ 10^{-07}$±$9.10\ 10^{-08}$ | 0.01±0.40 | 0.52±0.2 | 0.38±0.26 | $4.46\ 10^{-07}$±$9.14\ 10^{-08}$ | 0.00±0.05 | 0.46±0.22 |
| **Hybrids** | | | | | | | | | |
| $\sigma^2_Y$ | 68.70±97.56*** | 205.55±291.01*** | 2.30±3.30** | 0.004±001 ns | 1.25±1.89* | 1.91±2.92* | 171.38±278.12* | 43916.29±62145.09 ** | 1.95±11.14 ns |
| $\sigma^2_G$ | 7.99±3.23** | 0.72±1.79 ns | 0.4±0.16** | 0.00±0.01 ns | 0.23±0.3 ns | 0.00±0.00 ns | 271.11±303.23 ns | 127.57±303.17 ns | 82.04±70.95 ns |
| $\sigma^2_{GCA-F}$ | 5.19±5.16 ns | 1.29±1.61 ns | 0.15±0.09* | 0.01±0.01 ns | 0.40±0.36 ns | 0.00±0.00 ns | 91.94±221.98 ns | 37.53±89.05 ns | 56.66±99.03 ns |
| $\sigma^2_{GCA-M}$ | 2.38±1.84 ns | 1.41±1.2 ns | 0.33±0.18** | 0.00±0.00 ns | 0.00±0.00 ns | 0.10±0.2 ns | 0.00±0.00 ns | 39.56±210.96 ns | 7.82±15.71 ns |
| $\sigma^2_{SCA-F\times M}$ | 0.36±1.66 ns | 0.00±0.00 ns | 0.00±0.00 ns | 0.00±0.00 ns | 0.00±0.00 ns | 0.00±0.00 ns | 176.06±280.54 ns | 66.75±301.52 ns | 12.61±49.63 ns |
| $\sigma^2_{G\times Y}$ | 25.67±3.4*** | 18.32±2.45*** | 1.28±0.17*** | 0.11±0.01*** | 2.74±0.39*** | 8.34±0.85*** | 2793.87±390.55*** | 3113.3±414.17*** | 675.54±89.13*** |
| $\sigma^2_{GCA-F\times Y}$ | 9.33±4.69*** | 2.37±1.66** | 0.00±0.00 ns | 0.01±0.01* | 0.51±0.3*** | 2.52±0.99*** | 338.23±262.33* | 0.00±0.00 ns | 216.40±113.59*** |
| $\sigma^2_{GCA-M\times Y}$ | 1.79±1.46* | 0.87±1.05 ns | 0.09±0.09 ns | 0.00±0.01 ns | 0.43±0.19*** | 0.00±0.00 ns | 66.14±110.98 ns | 300.22±275.53 ns | 0.00±0.00 ns |
| $\sigma^2_{SCA-F\times M\times Y}$ | 15.36±2.29*** | 13.77±1.48*** | 1.10±0.11*** | 0.09±0.01*** | 1.70±0.2*** | 6.13±0.69*** | 2438.36±376.08*** | 2854.64±406.01*** | 475.39±66.14*** |
| $\sigma^2_e$ | 0.66±0.06 | 0.82±0.09 | 0.05±0.00 | 0.00±0.00 | 0.42±0.04 | 1.62±0.16 | 339.81±32.14 | 92.86±11.47 | 12.30±1.20 |
| $H^2$ | 0.55±0.21 | 0.38±0.19 | 0.61±0.11 | 0.32±0.20 | 0.36±0.22 | 0.04±0.08 | 0.37±0.32 | 0.21±0.47 | 0.34±0.36 |

$\sigma^2_G$: genotypic variance; $\sigma^2_Y$: year variance; $\sigma^2_{G\times Y}$: genotype×year interaction variance; $\sigma^2_e$, residual variance; $\sigma^2_{GCA-F}$: female general combining ability variance; $\sigma^2_{GCA-M}$: male general combining ability variance; $\sigma^2_{SCA-F\times M}$: specific combining ability variance; $\sigma^2_{GCA-F\times Y}$: female general combining ability×year interaction variance; $\sigma^2_{GCA-M\times Y}$: male general combining ability×year interaction variance; $\sigma^2_{SCA-F\times M\times Y}$: specific combining ability×year interaction variance; $H^2$: broad-sense heritability; ***, **, *: significantly different from zero at the 0.001, 0.01, and 0.05 probability levels, respectively. ns: not significantly different from zero at the 0.05 level of probability.

**Table 4. Estimates of genetic variance components, heritability, degree of dominance, predictability ratio, and genetic advance for the macroelements in 118 hybrids and their 26 parents of *Gynandropsis gynandra* evaluated in 2019 and 2020.**

| Source | | Ca | | K | | Mg | | P | | Na | |
|---|---|---|---|---|---|---|---|---|---|---|---|
| **Year** | | 2019 | 2020 | 2019 | 2020 | 2019 | 2020 | 2019 | 2020 | 2019 | 2020 |
| **Parents** | | | | | | | | | | | |
| $\sigma^2_G$ | | 62.33±19*** | 18.48±5.44*** | 17.9±5.42*** | 8.46±3.38*** | 2.24±0.67*** | 1.55±0.45*** | 3.87±1.16*** | 3.42±0.98*** | 0.05±0.02*** | 0.04±0.01*** |
| $\sigma^2_e$ | | 4.11±1.21 | 1.5±0.42 | 0.97±0.28 | 6.21±1.72 | 0.08±0.02 | 0.08±0.02 | 0.1±0.03 | 0.09±0.03 | 0.01±0 | 0±0 |
| $H^2$ | | 0.97±0.01 | 0.96±0.02 | 0.97±0.01 | 0.73±0.11 | 0.98±0.01 | 0.98±0.01 | 0.99±0.01 | 0.99±0.01 | 0.91±0.04 | 0.96±0.01 |
| Mean | | 31.34±1.49 | 18.17±0.85 | 23.4±0.78 | 27.39±1.52 | 3.74±0.2 | 6.15±0.19 | 7.33±0.25 | 8.61±0.21 | 0.94±0.09 | 1.15±0.04 |
| GAM | | 51.06 | 47.77 | 36.76 | 18.71 | 81.75 | 41.16 | 54.91 | 43.96 | 46.26 | 34.58 |
| **Hybrids** | | | | | | | | | | | |
| $\sigma^2_G$ | | 58.38±7.66*** | 8.66±1.19*** | 9.23±1.23*** | 29.41±3.93*** | 2.53±0.33*** | 0.83±0.11*** | 4.21±0.56*** | 1.51±0.25*** | 0.08±0.01*** | 0.15±0.02*** |
| $\sigma^2_{GCA-F}$ | | 27.75±13.04*** | 1.4±0.88** | 0.73±0.66 ns | 6.6±3.82*** | 0.26±0.18** | 0.04±0.05 ns | 1.5±0.74*** | 0.28±0.19*** | 0.03±0.01*** | 0.01±0.01 ns |
| $\sigma^2_{GCA-M}$ | | 6.69±3.87*** | 1.78±0.99*** | 1.09±0.79* | 3.41±2.48* | 0.66±0.34*** | 0.21±0.11*** | 0.67±0.38*** | 0.19±0.15* | 0.01±0.01** | 0.00±0.01 ns |
| $\sigma^2_{SCA-F\times M}$ | | 25.68±3.79*** | 5.53±0.87*** | 7.52±1.12*** | 20.55±3.12*** | 1.62±0.24*** | 0.58±0.09*** | 2.16±0.33*** | 1.09±0.22*** | 0.04±0.01*** | 0.14±0.02*** |
| $\sigma^2_e$ | | 0.48±0.06 | 0.84±0.11 | 0.29±0.04 | 1.31±0.17 | 0.04±0.01 | 0.06±0.01 | 0.16±0.02 | 0.71±0.09 | 0.00±0.00 | 0.01±0.00 |
| $\sigma^2_A$ | | 68.87±27.11 | 6.36±2.61 | 3.64±2.09 | 20.03±9.12 | 1.83±0.76 | 0.49±0.24 | 4.34±1.66 | 0.93±0.48 | 0.08±0.03 | 0.03±0.03 |
| $\sigma^2_D$ | | 102.7±15.14 | 22.14±3.48 | 30.06±4.48 | 82.2±12.48 | 6.48±0.96 | 2.32±0.36 | 8.64±1.31 | 4.34±0.87 | 0.16±0.02 | 0.56±0.08 |
| $h^2$ | | 0.4±0.1 | 0.22±0.08 | 0.11±0.06 | 0.19±0.08 | 0.22±0.08 | 0.17±0.07 | 0.33±0.09 | 0.17±0.08 | 0.32±0.1 | 0.05±0.05 |
| $H^2$ | | 0.99±0.01 | 0.95±0.01 | 0.98±0.00 | 0.98±0.00 | 0.99±0.00 | 0.97±0.01 | 0.98±0.00 | 0.81±0.04 | 0.99±0.00 | 0.98±0.00 |
| $h^2_F$ | | 0.65±0.22 | 0.19±0.12 | 0.09±0.08 | 0.26±0.14 | 0.12±0.09 | 0.06±0.06 | 0.46±0.18 | 0.20±0.13 | 0.5±0.19 | 0.09±0.08 |
| $h^2_M$ | | 0.16±0.09 | 0.25±0.13 | 0.13±0.09 | 0.13±0.09 | 0.32±0.14 | 0.29±0.14 | 0.20±0.11 | 0.13±0.10 | 0.14±0.09 | 0.01±0.05 |
| Degree of dominance | | 1.73±0.37 | 2.64±0.61 | 4.07±1.27 | 2.86±0.72 | 2.66±0.61 | 3.08±0.81 | 1.99±0.42 | 3.05±0.9 | 2.07±0.46 | 6.18±3.23 |
| Predictability ratio | | 0.57±0.1 | 0.36±0.11 | 0.19±0.10 | 0.33±0.11 | 0.36±0.11 | 0.3±0.11 | 0.5±0.11 | 0.3±0.12 | 0.48±0.11 | 0.09±0.09 |
| Mean | | 29.32±0.49 | 17.57±0.74 | 24.1±0.38 | 44.39±0.89 | 3.85±0.15 | 6.01±0.27 | 8.73±0.28 | 10.36±0.65 | 0.84±0.03 | 0.97±0.09 |
| GAM | | 53.57 | 33.69 | 25.76 | 24.89 | 84.80 | 30.62 | 47.99 | 21.93 | 66.89 | 82.60 |

$\sigma^2_A$: additive genetic variance; $\sigma^2_D$: dominance genetic variance; $\sigma^2_{GCA-F}$: female general combining ability variance; $\sigma^2_{GCA-M}$: male general combining ability variance; $\sigma^2_{SCA-F\times M}$: specific combining ability variance; $\sigma^2_e$: residual variance; $\sigma^2_G$: genotypic variance; $H^2$: broad-sense heritability; $h^2$: narrow-sense heritability; $h^2_F$: narrow-sense heritability for females; $h^2_M$: narrow-sense heritability for males; GAM: genetic advance over mean. ***, **, *: significantly different from zero at the 0.001, 0.01, and 0.05 probability levels, respectively. ns: not significantly different from zero at the 0.05 level of probability.

**Table 5. Estimates of genetic variance components, heritability, degree of dominance, predictability ratio, and genetic advance for the microelements in 118 hybrids and their 26 parents of *Gynandropsis gynandra* evaluated in 2019 and 2020.**

| Source | Fe | | Cu | | Mn | | Zn | |
|---|---|---|---|---|---|---|---|---|
| Year | 2019 | 2020 | 2019 | 2020 | 2019 | 2020 | 2019 | 2020 |
| **Parents** | | | | | | | | |
| $\sigma^2_G$ | 1988.93±702.42*** | 3925.55±1273.13*** | 8.9±2.79*** | 4.2±2.26* | 16.13±6.39*** | 8467.17±2401.96*** | 111.33±34.31*** | 238.41±69.04*** |
| $\sigma^2_e$ | 731.98±211.3 | 1087.85±301.72 | 1.12±0.32 | 6.26±1.77 | 9.95±2.87 | 50.04±13.88 | 9.87±2.85 | 11.27±3.19 |
| $H^2$ | 0.84±0.06 | 0.88±0.05 | 0.94±0.02 | 0.57±0.17 | 0.76±0.1 | 1±0 | 0.96±0.02 | 0.98±0.01 |
| Mean | 183.37±17.65 | 135.07±21.92 | 9.1±0.73 | 12.04±1.5 | 26.74±1.96 | 186.64±4.99 | 51.16±2.18 | 60.61±2.42 |
| GAM | 46.04 | 89.55 | 65.52 | 26.55 | 27.06 | 101.41 | 41.58 | 51.87 |
| **Hybrids** | | | | | | | | |
| $\sigma^2_G$ | 3584.63±498.46*** | 2574.02±354.39*** | 10.72±1.56*** | 6.33±0.89*** | 345.86±46.08*** | 6308.96±839.47*** | 1232.78±162.24*** | 287.46±38.13*** |
| $\sigma^2_{GCA-F}$ | 909.69±524.4*** | 11.78±120.12 ns | 5.16±2.53*** | 0.01±0.32 ns | 46.63±34.55* | 3.16±275.78 ns | 502.26±246.67*** | 44.01±29.3** |
| $\sigma^2_{GCA-M}$ | 15.77±149.21 ns | 97.22±159.77 ns | 0.16±0.39 ns | 0.05±0.37 ns | 0.00±0.00 ns | 690.07±594.96 ns | 0.00±0.00 ns | 19.7±18.81 ns |
| $\sigma^2_{SCA-F\times M}$ | 2757.39±433.37*** | 2470.52±378.24*** | 5.98±1.04*** | 6.27±0.99*** | 305.27±42.92*** | 5726.51±859.67*** | 746.7±103.64*** | 228.43±33.98*** |
| $\sigma^2_e$ | 412.01±53.87 | 266.54±34.7 | 2.29±0.3 | 0.96±0.13 | 13.02±1.7 | 167.4±21.98 | 16.23±2.12 | 8.28±1.08 |
| $\sigma^2_A$ | 1850.91±1088.4 | 218.01±412.88 | 10.65±5.12 | 0.11±1 | 93.27±69.09 | 1386.47±1325.99 | 1004.52±493.34 | 127.42±70.55 |
| $\sigma^2_D$ | 11029.58±1733.49 | 9882.07±1512.96 | 23.91±4.18 | 25.09±3.97 | 1221.06±171.67 | 22906.04±3438.7 | 2986.8±414.57 | 913.71±135.9 |
| $h^2$ | 0.14±0.08 | 0.02±0.04 | 0.3±0.11 | 0±0.04 | 0.07±0.05 | 0.06±0.05 | 0.25±0.1 | 0.12±0.06 |
| $H^2$ | 0.95±0.01 | 0.95±0.01 | 0.90±0.02 | 0.93±0.01 | 0.98±0.00 | 0.99±0.00 | 0.99±0.00 | 0.99±0.00 |
| $h^2_F$ | 0.28±0.15 | 0±0.05 | 0.58±0.21 | 0±0.05 | 0.14±0.1 | 0±0.05 | 0.5±0.19 | 0.17±0.11 |
| $h^2_M$ | 0±0.05 | 0.04±0.06 | 0.02±0.04 | 0.01±0.06 | 0.00±0.00 | 0.11±0.1 | 0.00±0.00 | 0.08±0.07 |
| Degree of dominance | 3.45±1.09 | 9.52±9.31 | 2.12±0.56 | 20.98±92.83 | 5.12±1.98 | 5.75±2.91 | 2.44±0.63 | 3.79±1.14 |
| Predictability ratio | 0.25±0.12 | 0.04±0.08 | 0.47±0.13 | 0.01±0.08 | 0.13±0.09 | 0.11±0.1 | 0.4±0.12 | 0.22±0.1 |
| Mean | 161.44±14.35 | 141.69±11.26 | 10.68±1.02 | 8.65±0.76 | 44.43±2.53 | 340.89±9.72 | 75.17±2.86 | 71.21±2.06 |
| GAM | 74.29 | 71.92 | 60.03 | 57.75 | 85.43 | 47.68 | 95.90 | 48.70 |

$\sigma^2_A$: additive genetic variance; $\sigma^2_D$: dominance genetic variance; $\sigma^2_{GCA-F}$: female general combining ability variance; $\sigma^2_{GCA-M}$: male general combining ability variance; $\sigma^2_{SCA-F\times M}$: specific combining ability variance; $\sigma^2_e$: residual variance; $\sigma^2_G$: genotypic variance; $H^2$: broad-sense heritability; $h^2$: narrow-sense heritability; $h^2_F$: narrow-sense heritability for females; $h^2_M$: narrow-sense heritability for males; GAM: genetic advance over mean. ***, **, *: significantly different from zero at the 0.001, 0.01, and 0.05 probability levels, respectively. ns: not significantly different from zero at the 0.05 level of probability.

## General combining ability effects of the parents

Estimates of the general combining ability (GCA) effects of male and female parents are presented in Fig 2. Female parents had significant GCA effects for all minerals, while no significant GCA effects were observed for male parents for Zn, Fe, Mn, and Cu in both years. Some female and male parents displayed multiple significant and positive GCA effects. Good general female combiners included parent P12 for Ca, Mg and Cu; parent P09 for K, Na, Fe, Cu and Mn; P11 for Ca, Mg, K and Fe; parent P10 for Ca, K, Mg, Na and Cu; parent P05 for Zn and P. Parent P01 was good combiner for Zn. Males with multiple positive GCA effects comprised parent P25 for Ca, Mg, Na, and Mn; parents P26 and P15 for Ca and Mg, and P24 for P and K.

## Specific combining ability effects of hybrids

A wide range of specific combining ability (SCA) effects from negative to positive was observed for all mineral content (S2–S10 Fig). The hybrids P10×P13, P10×P14, P07×P20, P02×P15 and P05×P25 exhibited highly significant and positive SCA effects for Ca (S2 Fig). Similarly, significant and positive SCA effects for Mg in both years were observed

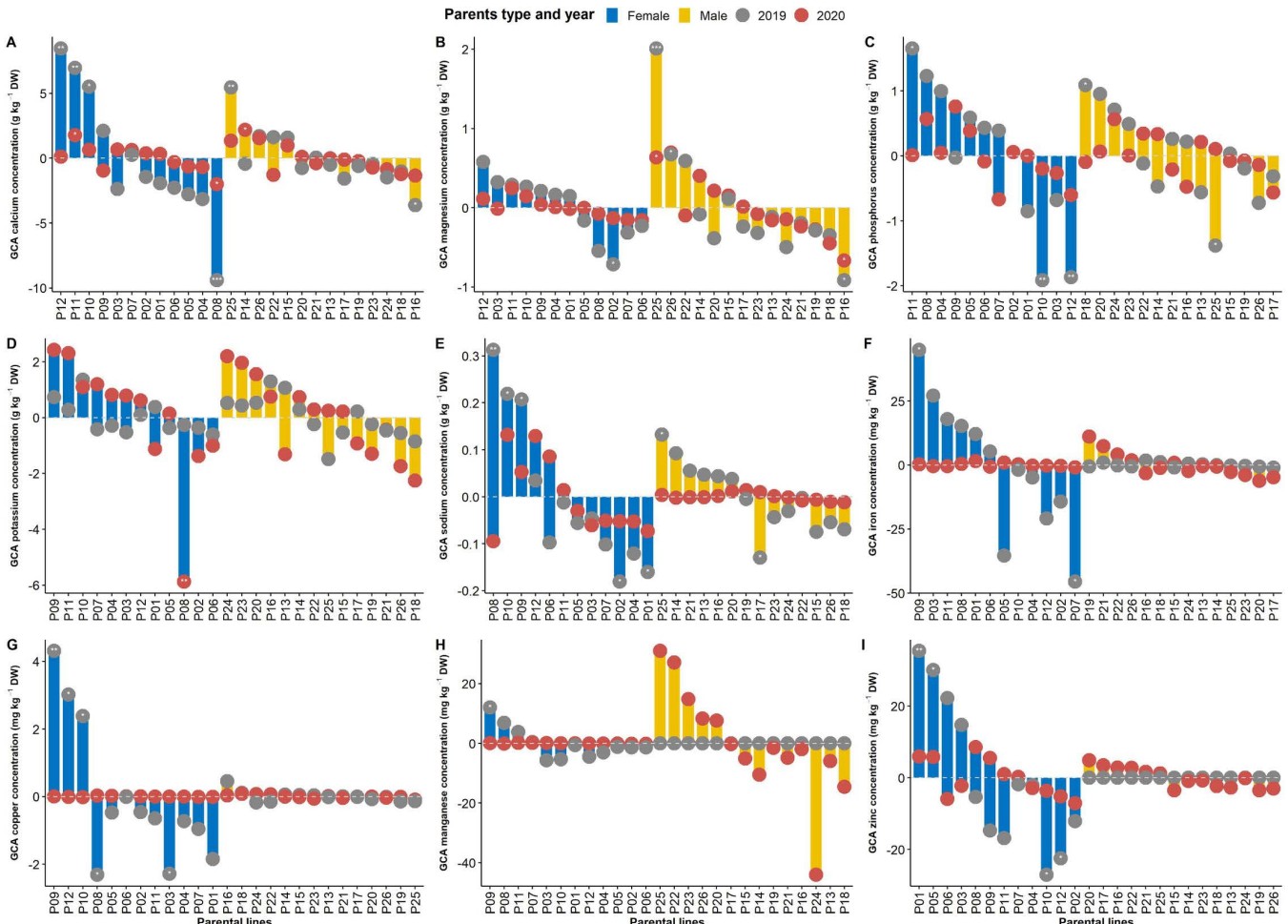

**Fig 2. Estimates of general combining ability effects of male and female lines for mineral elements involved in 118 experimental hybrids of *Gynandropsis gynandra* evaluated in 2019 and 2020.** \*\*\*, \*\*, \* refer to estimate of general combining ability effect significantly different from zero at p<0.001, 0.01, and 0.05, respectively.

for crosses P12×P26, P10×P13, P04×P17, P02×P15, and P10×P14 (S3 Fig). In contrast, negative desirable SCA effects were observed in crosses P10×P17, P05×P13, P03×P23, P08×P18 and P08×P15 for Ca, and P08×P18, P02×P13, P10×P17, P05×P13, and P08×P19 for Mg. As phosphorus concerned, crosses with positive and significant SCA effects were P05×P15, P02×P24, P03×P24, P09×P19, P08×P22, and P08×P15 and hybrids with negative and significant SCA effects included P09×P16, P07×P17, P07×P21 and P07×P26 (S4 Fig). While the hybrids P10×P18, P11×P16, P12×P25, P10×P17, P09×P24 had significant and positive SCA effects for potassium content, P08×P16, P01×P25, P02×P18, P05×P13, P08×P19 had a negative and significant SCA effects (S5 Fig). The crosses such as P10×P16, P09×P19, P03×P25, P12×P25, P11×P20 and P02×P20 displayed highly significant and positive SCA effects for sodium. In contrast, progenies of P05×P24, P01×P26, P04×P15, P08×P18, P06×P18 showed negative and significant SCA effects (S6 Fig). For Fe, the highest and most significant SCA effects were observed in hybrid P01×P21, followed by P09×P19, P05×P24, P08×P22 and P10×P25. Negative and significant SCA effects were displayed by hybrids P06×P21, P09×P24, P07×P17 and P10×P20 for Fe content (S7 Fig). Hybrids P02×P16, P09×P19, P12×P16, P08×P16, P02×P14 and P12×P13 were the best for SCA effects, while P11×P25, P04×P25, P12×P26, P09×P15 had negative and significant SCA effects for copper content (S8 Fig). For Mn content, while crosses with significant and positive SCA effects included P09×P23, P12×P13, P09×P21, P07×P25, P11×P22 and P11×P19, the ones with negative SCA effects included P09×P15, P06×P13, P06×P26, P04×P20, P04×P24 and P02×P13 (S9 Fig). The highest and significantly positive SCA effect for Zn was observed in P01×P21, followed by P06×P16, P09×P18, P11×P16, and P02×P16, whereas crosses such as P01×P16, followed by P09×P15, P02×P25, P11×P16, and P05×P19 had negative desirable SCA effects (S10 Fig).

## Heterosis

The distributions of mid- and best-parent heterosis (MPH and BPH, respectively) for mineral elements are shown in Fig 3. A similar distribution pattern was observed for most traits each year. The heterosis for mineral content ranged between −80.4% and 389.5%, respectively, when mid- and best-parent heterosis were pooled. The species displayed both negative and positive heterosis. For Ca, the hybrids with positive BPH and MPH were P07×P20, P09×P20, P09×P21, and P10×P14 across years (S11 and S12 Figs). While the hybrids P04×P17, P07×P20, P04×P18, P12×P25, P10×P13 and P10×P14 exhibited positive MBP and BPH for magnesium (S13 and S14 Figs), the crosses P04×P25, P11×P25, P11×P13, P05×P14, P02×P14, P08×P15 had positive mid- and best parents heterosis for phosphorus across years (S15 and S16 Figs). For potassium, P01×P13, P02×P14, P10×P16, P08×P15 and P10×P20 were the best (S17 and S18 Figs) and P09×P19, P10×P16 and P12×P25 were identified for sodium (S19 and S20 Figs). The top five crosses with positive mid- and best parents heterosis in both years for iron included P01×P21, P06×P16, P09×P19, P10×P25 and P11×P15 (S21 and S22 Figs). Regarding copper, crosses P04×P15 and P08×P17 were the best (S23 and S24 Figs). The cross combinations P02×P15, P04×P15, P07×P15, P07×P20, P09×P21, and P09×P23 were the best with positive MPH and BPH for manganese (S25 and S26 Figs). For zinc, the hybrids P01×P21, P01×P23, P05×P14, P06×P17, P01×P17, P04×P18, P05×P25 displayed positive MPH and BPH (> 30%) across years (S27 and S28 Figs).

## Correlation between combining ability, heterosis and mean performance of the parents and hybrids

The associations between combining ability, heterosis, and mean genotypic values of the parents and hybrids are summarized in Table 6. The per se performance of the parents exhibited a significantly positive correlation with their GCA effects for calcium, potassium, magnesium, and zinc in 2020 only. The $F_1$ per se performance was significantly and positively correlated with SCA and the sum of GCA effects (sGCA) of the hybrids' parents for all minerals (Table 6). The r (SCA, $F_1$) was higher than the r ($F_1$, sGCA) for all traits. The correlation coefficients between SCA and $F_1$ were strong (r ≥ 0.70). In addition, heterosis (both MPH and BPH) showed a highly significant and positive association with SCA and $F_1$ hybrid performance for all minerals, ranging from moderate to strong and trait specific. The correlation between the sum of GCA

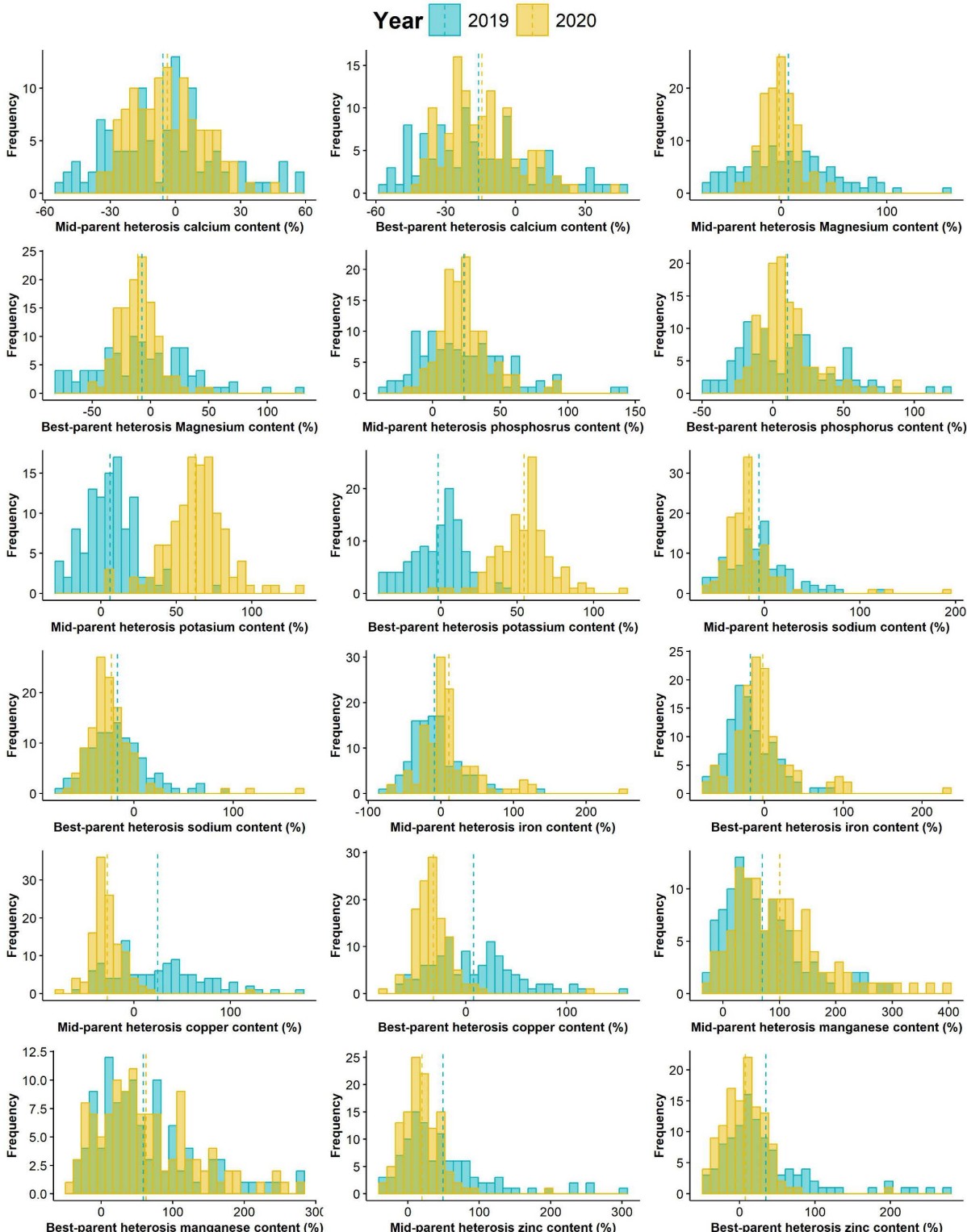

**Fig 3. Distribution of mid- and best-parent heterosis for nine mineral elements in 118 experimental hybrids of *Gynandropsis gynandra* evaluated in 2019 and 2020.** A dotted line indicates the mean of each year.

**Table 6. Spearman correlation coefficients between general combining ability effects and parent *per se* performance r(*per se*, GCA), among mid- (MPH) and best- (BPH) parent heterosis, hybrid performance (F$_1$) and specific combining ability (SCA) and sum of general combining abilities of hybrids' parents (sGCA) for nine leaf mineral content in *G. gynandra*.**

| Traits | Year | r(*per se*, GCA) | r(F$_1$, SCA) | r(F$_1$, MPH) | r(F$_1$, BPH) | r(SCA, MPH) | r(SCA, BPH) | r(F$_1$, sGCA) | r(sGCA, MPH) | r(sGCA, BPH) |
|---|---|---|---|---|---|---|---|---|---|---|
| **Ca** | 2019 | 0.21ns | 0.70*** | 0.75*** | 0.71*** | 0.54*** | 0.49*** | 0.72*** | 0.52*** | 0.50*** |
| | 2020 | 0.68*** | 0.83*** | 0.54*** | 0.62*** | 0.71*** | 0.70*** | 0.69*** | 0.03ns | 0.20* |
| **Cu** | 2019 | −0.07 ns | 0.75*** | 0.85*** | 0.85*** | 0.52*** | 0.50*** | 0.69*** | 0.77*** | 0.75*** |
| | 2020 | 0.22ns | 0.99*** | 0.62*** | 0.60*** | 0.89*** | 0.83*** | 0.41*** | 0.35*** | 0.37*** |
| **Fe** | 2019 | −0.04 ns | 0.80*** | 0.87*** | 0.78*** | 0.71*** | 0.66*** | 0.63*** | 0.57*** | 0.52*** |
| | 2020 | 0.16ns | 0.98*** | 0.89*** | 0.83*** | 0.65*** | 0.65*** | 0.40*** | 0.09ns | 0.02ns |
| **K** | 2019 | 0.26ns | 0.93*** | 0.67*** | 0.65*** | 0.65*** | 0.63*** | 0.61*** | 0.32*** | 0.33*** |
| | 2020 | 0.51** | 0.88*** | 0.81*** | 0.83*** | 0.79*** | 0.78*** | 0.58*** | 0.33*** | 0.41*** |
| **Mg** | 2019 | 0.14ns | 0.85*** | 0.86*** | 0.83*** | 0.77*** | 0.75*** | 0.64*** | 0.53*** | 0.52*** |
| | 2020 | 0.54** | 0.88*** | 0.52*** | 0.52*** | 0.66*** | 0.60*** | 0.65*** | 0.06ns | 0.14ns |
| **Mn** | 2019 | 0.29ns | 0.96*** | 0.97*** | 0.97*** | 0.96*** | 0.95*** | 0.41*** | 0.35*** | 0.34*** |
| | 2020 | 0.22ns | 0.98*** | 0.50*** | 0.50*** | 0.52*** | 0.52*** | 0.39*** | 0.09ns | 0.13ns |
| **Na** | 2019 | 0.03ns | 0.75*** | 0.86*** | 0.81*** | 0.67*** | 0.63*** | 0.73*** | 0.63*** | 0.62*** |
| | 2020 | 0.34ns | 0.91*** | 0.84*** | 0.78*** | 0.83*** | 0.79*** | 0.45*** | 0.30*** | 0.25** |
| **P** | 2019 | 0.16ns | 0.76*** | 0.75*** | 0.64*** | 0.64*** | 0.56*** | 0.78*** | 0.53*** | 0.46*** |
| | 2020 | 0.35ns | 0.93*** | 0.43*** | 0.48*** | 0.50*** | 0.47*** | 0.57*** | 0.02ns | 0.21* |
| **Zn** | 2019 | 0.32ns | 0.69*** | 0.90*** | 0.88*** | 0.71*** | 0.70*** | 0.76*** | 0.61*** | 0.59*** |
| | 2020 | 0.58** | 0.88*** | 0.60*** | 0.52*** | 0.68*** | 0.59*** | 0.62*** | 0.13ns | 0.11ns |

Ca, K, Mg and Na (g kg$^{-1}$); Cu, Fe, Mn and Zn (mg kg$^{-1}$); ***, **, * = significantly different from zero at the 0.001, 0.01, and 0.05 probability levels, respectively. ns = not significantly different from zero at the 0.05 level of probability.

effects of the parents of hybrids and heterosis was variable and depended on the trait and year. In 2019, the sum of GCA effects of hybrid parents had a moderate and positive correlation with MPH and BPH for all minerals. In contrast, no weak correlations were observed between the sum of the GCA hybrid parents and the heterosis in 2020 (Table 6).

### Correlation of parents' performance, hybrid phenotype, specific combining ability effects, and heterosis among leaf mineral elements

Variable and significant correlations among the leaf mineral contents for both parents and hybrids were observed, and the correlations ranged from weak to strong (Figs 4A–D). Despite some changes observed from 2019 to 2020 and from parents to hybrids, some leaf mineral content displayed similar correlation patterns in both years. Calcium and magnesium contents had a significant, positive, and strong correlation for both parents and hybrids (r ≥ 0.68, p < 0.001). A positive and significant association was observed between iron and phosphorus contents for both parents and hybrids. There was a moderate, positive, and significant correlation between phosphorus and copper contents (r ≥ 0.43, p < 0.05) for the parents. For hybrids, sodium content had a positive and significant correlation with potassium and magnesium contents. In addition, zinc content had a moderate and positive correlation with phosphorus content (r = 0.31, p < 0.01).

Significant and positive correlations were depicted among the specific combining ability (SCA) effects of the leaf mineral elements (Figs 4E–F). The SCA effect of magnesium had a strong, positive, and significant correlation with the SCA effect of calcium (r ≥ 0.66, p < 0.001). There was a positive and significant correlation between the SCA effects of iron and zinc (r ≥ 0.22, p < 0.05). Additionally, the SCA effect of iron had a positive and significant association with that of phosphorus (r ≥ 0.22, p < 0.05) and copper (r = 0.35, p < 0.001). Similarly, the SCA effect of zinc had a positive and significant

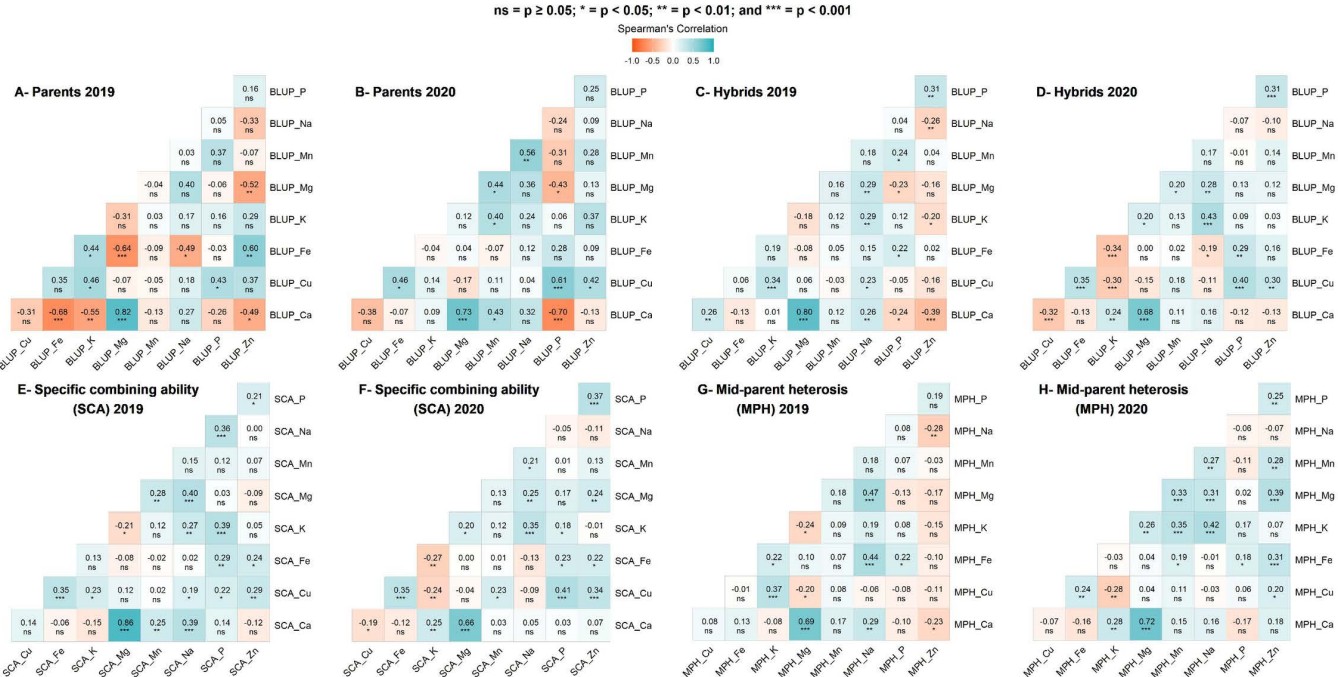

**Fig 4. Correlations among leaf mineral elements based on the phenotypic performance for the 26 parental lines in 2019 (A) and 2020 (B), the phenotypic performance for the 118 hybrids in 2019 (C) and 2020 (D), specific combining ability effects for the 118 hybrids in 2019 (E) and 2020 (F) and mid-parent heterosis for the 118 hybrids in 2019 (G) and 2020 (H).**

association with the SCA effects of phosphorus (r ≥ 0.21, p < 0.05) and copper (r = 0.29, p < 0.01). There were significant and positive correlations between SCA effects of sodium and potassium (r ≥ 0.27, p < 0.01), copper and phosphorus (r ≥ 0.22, p < 0.05), magnesium and sodium (r ≥ 0.25, p < 0.01), and potassium and phosphorus (r ≥ 0.18, p < 0.05).

Overall, the correlations among heterosis in leaf mineral content varied significantly (Figs 4 G and H). We observed a significant, strong, and positive correlation between heterosis of calcium and magnesium (r ≥ 0.69, p < 0.001). There was also a positive, moderate, and significant correlation between the heterosis of magnesium and sodium (r ≥ 0.31, p < 0.001). Heterosis of iron had a positive and significant correlation with phosphorus but was weak (r < 0.30, p < 0.05).

## Discussion

In the present study, we observed highly significant variation among parents and $F_1$ hybrids for leaf mineral content, showing the existence of functional genetic variation for spider plant improvement. The variation could be associated with the diversity in the origin of lines used in the present study, in line with reports by Sogbohossou et al. [11], Blalogoe et al. [62], and Reeves et al. [22]. Moreover, as observed in this study, Omondi et al. [12] reported significant variation in leaf mineral content among genotypes composed of farmers' cultivars, advanced lines, and germplasm accessions from Eastern and Southern Africa. Similar observations for leaf mineral content were observed among Southern and Eastern genotypes [63,64] as well as among worldwide accessions in seed mineral composition [62], secondary metabolite concentrations [16], leaf vitamins content and morphological traits [11] and antioxidant activity [20]. More importantly, the genetic makeup of the genotypes has been reported to be associated with the phenotypic variation in *G. gynandra* [19,27].

Overall, hybrids outperformed their parents in several mineral contents, including zinc, potassium, phosphorus, and manganese, indicating hybrid vigor. Similar results were reported for mineral elements in *Brassica oleracea* var. *capitata*

[38], in *Brassica rapa* [39], and in *Brassica oleracea* var. *botrytis* L. [40,41]. In addition, some hybrids were better or worse than their specific parents for various traits, showing that different mechanisms control the inheritance of the characteristics. The average performance of several hybrids for mineral content was higher than that of the parents and accessions previously reported by [12,14,63–66]. For instance, the average zinc content of 60 mg kg$^{-1}$ dry weight (DW) with a maximum of 80 mg kg$^{-1}$ DW reported by Omondi et al. [12] was lower than that of the present study, which was 73.19 mg kg$^{-1}$ DW with a maximum of 224.95 mg kg$^{-1}$ DW. The average Ca content in hybrids was twofold higher than that reported by Jiménez-Aguilar and Grusak [64]. However, we observed lower values for iron content than those of Moenga et al. [67], Omondi et al. [12], Gowele et al. [15], and Thovhogi et al. [63]. This might be due to the genotypes, the leaves sampling stage, cultivation practices, and growth conditions. For instance, Jinazali et al. [66] observed significant variation in mineral content between genotypes collected from different agroecological areas of Malawi. At the same time, Mamboleo et al. [68] and Makokha et al. [69] reported the effect of harvesting stage and growing locations on the leaf mineral content in *G. gynandra*. Further investigations are therefore required to assess the phenotypic plasticity and stability of the newly developed hybrids, identifying the best genotypes under various agroecological and agronomic practices in field conditions.

Knowledge of gene action and combining ability effects of traits is important for any crop breeding. Gene action is particularly crucial in the choice of breeding method [70]. Genetic variance component analysis revealed that both additive and non-additive genes influenced all traits. Nonadditive gene effects were predominant for all leaf mineral elements (Ca, Mg, K, P, Na, Fe, Zn, Cu, and Mn) in *G. gynandra*. This finding was supported by (1) the specific combining ability effects and dominance variances that were greater than the general combining ability effects and additive variances, respectively; (2) the degree of dominance greater than unity; and (3) the predictability ratio below 0.5 for all mineral contents. Consequently, the selection for nutritious cultivars in *G. gynandra* should focus on the development of hybrid cultivars for better exploitation of the dominance gene action through recurrent selection, especially reciprocal recurrent selection [70]. Similar findings were obtained by Xie et al. [39] for Ca, Fe, Mg, and Zn in *Brassica rapa* (Chinese cabbage), Singh et al. [71] for Fe, Zn, Cu, Mn, K and Ca in *Brassica oleracea* var. *capitata* (cabbage) where nonadditive gene action was predominant. Assessing epistasis was not part of the objectives of this study. Further studies using appropriate mating designs, such as North Carolina Design III and triple test cross should be implemented to estimate its contribution to the inheritance of mineral content in the species.

Heritability is a key parameter in breeding, particularly in the prediction of the response to selection. Broad-sense heritability ($H^2$) is a measure of the proportion of the total phenotypic variation attributable to the variance of genetic values [72]. Broad-sense heritability estimates were high (> 0.60) for all mineral contents per year, showing that phenotypic variation observed among genotypes is primarily due to genotypic variation. Similar results were found concerning mineral content in the species [17] and *Amaranthus tricolor* L. [73]. Furthermore, narrow-sense heritability ($h^2$) is a measure of the proportion of the total phenotypic variation attributable to additive genetic variance. A high $h^2$ value indicates that phenotypic variation is primarily due to additive genetic effects. Thus, the higher the $h^2$ value is, the better the response to selection will be. We observed low to moderate narrow-sense heritability for all mineral contents (< 50%), which agreed with the preponderance of dominance genes in the inheritance of mineral content in the species. Xie et al. [39] also reported low and moderate $h^2$ for mineral content in *Brassica rapa*. In contrast, Karmakar et al. [74] reported low $h^2$ for mineral and antioxidant content in ridge gourd (*Luffa acutangula* Roxb.).

Moreover, the genotype × year interaction was significant, with variance greater than the genotypic variance. This indicates phenotypic plasticity in the species. Similarly, considerable genotype by environment interaction effect was reported on leaf mineral content in the species [17], switchgrass tiller mineral content [75], rice seed elemental composition [76], maize kernel mineral concentration [77], seed iron and zinc content of sorghum [78] and common bean [79]. Moreover, the plant mineral composition is influenced by the crop genetics, the growing environment, the management practices, and their interaction [80,81]. The environment encompasses various factors, including soil conditions and climate, such

as relative humidity, light intensity, and temperature, among others. Potential contributing factors to the observed genotype × year interaction might include soil type, temperature, and relative humidity. Soil could be the main driving factor, as plants absorb nutrients from the soil, and different types were used in this study, with soil in 2019 and growing media in 2020. However, this requires further investigation by evaluating these parents and progenies in multiple environments to confirm the level of environmental influence and genotype-by-environment interactions in phenotypic variation, and to identify key loci controlling leaf mineral content through ecological interaction, as reported in various species, including rice, maize, and switchgrass [75–77].

The expected genetic gain, another important metric for breeding for quantitative traits, estimates the quantity of increase in performance between the selected and base populations and is key, along with heritability, in any breeding program [82]. Our expected genetic gain at a selection intensity of 5% for all the mineral content ranged between 21.93% and 95.90%. A genetic gain of over 20% was observed for all minerals, indicating that significant improvements would be achieved through selection.

General combining ability is crucial for parental selection, while specific combining ability is essential for optimal cross selection to harness heterosis. Although nonadditive gene effects were predominant, high and significant general combining ability effects were observed for some female and/or male parents in terms of mineral contents, and these effects were mainly due to additive and additive × additive gene effects [83]. The parents with good GCA effects are excellent founders for the development of improved populations and could be exploited through several generations of hybridization. In the present study, neither male nor female parents simultaneously showed significant GCA effects in the desired direction for all mineral contents. This result concurs with previous findings concerning minerals in cabbage head [71] and minerals, vitamins and antioxidants in cauliflower [40,83]. However, some females (P12, P09, P11, P10, P05) and males (P25, P26, P15, P24) parents are suitable for their multiple positive and significant GCA effects. These parents are excellent and valuable candidates and resources for developing improved populations for research and breeding purposes.

Specific combining ability effects result from nonadditive gene effects, comprising dominance and epistasis [34]. In this study, none of the crosses displayed high and significant SCA effects for all minerals. This finding is similar to the results of Parkash et al. [84] for antioxidant compounds in *Brassica oleracea* var. *capitata*. Still, it contrasts with those of Singh et al. [71] and Xie et al. [39]. The latter researchers were able to identify at least one cross with significant and positive SCA effects for all the investigated minerals in cabbage head and non-heading Chinese cabbage. However, depending on the targeted mineral, hybrids with the highest and most significant SCA in the desirable direction involved (i) both parents with good and significant GCA effects (e.g., P12 × P25 for Mg, P09 × P24 for K, P09 × P19 for Fe); (ii) one good and one poor combiner (e.g., P05 × P25 for Ca, P09 × P21 for Mn, P10 × P18 for K); and (iii) both parents with medium or bad GCA effects (e.g., P12 × P13 for Mn, P02 × P15 for Mg, P09 × P18 for Zn). This finding shows that depending on the trait, the observed SCA effect might result from (i) the cumulative effects of additive genes (good x good parents); (ii) the interaction between additive and nonadditive genes (good x poor general combiners or vice versa); and (iii) the over manifestation of the interaction between nonadditive genes, especially complementary epistatic effects [34,39,71,83,85]. We also observed that some crosses involving both parents with good GCA displayed significant and negative SCA effects. This might be the result of the absence of or weak interaction between the desirable alleles. Therefore, crosses from good general combiners might not always display desirable SCA effects. Based on the above, breeding strategies for high-quality leaves in *G. gynandra* should consider both GCA and SCA in the selection of superior parents and crosses. Heterosis breeding and recurrent selection, along with multiple crossing programs, can be implemented. Types of cultivars may include hybrids, synthetics, composites, and population improvements. Strategies implemented for allogamous species can also be applied to this species. Therefore, breeding strategies should focus on (i) selecting parents with good general combining ability, followed by (ii) a selection based on specific combining ability. Reciprocal recurrent selection would be the most effective method for exploiting both additive and non-additive gene action in the species.

Heterosis, also known as hybrid vigor, refers to the outperformance of F1 progeny over their parents and has significantly contributed to increased crop productivity. Here, we report for the first time this phenomenon in *G. gynandra*. The level of heterosis over the mid and best parents was significant and variable between leaf mineral concentrations. This agrees with earlier reports on the existence of heterosis for mineral content, vitamins, antioxidants, and proteins in vegetable crops such as cabbage, tomato, cauliflower, pea and bean [38,39,83,86–90]. Specifically, the level of heterosis observed in the present study was higher than that reported for minerals in non-heading Chinese cabbage [39] and *Brassica oleracea* var. *capitata* [38]. The result was comparable to the level of heterosis for vitamins and antioxidant pigments in cauliflower [83] and cabbage [84] and some bioactive properties in interspecific crosses between cultivated and wild relatives of eggplant [91]. The wide range of heterosis could be explained by the previous observations on the reproductive biology of the species, revealing that the species is predominantly outcrossing [25–27].

Both negative and positive mid- and best parent heterosis were observed in the species for all mineral content. This might be because several mechanisms are underlying heterosis expression in *G. gynandra*. Three main models have been widely used to explain heterosis in crops, including dominance, overdominance, and epistasis [29,92–94]. Moreover, hybrids exhibiting a high level of heterosis are a combination of parents with either both good, both poor, average x good, good x poor, average x average, or average x poor general combining abilities. The results showed that all three models or their combination could explain heterosis in *G. gynandra,* as most research has highlighted that a single model rarely occurs in plants [92,95]. The present observation of the existence of heterosis in *G. gynandra* adds to previous reports [7,96] that the species could be used as a model for heterosis studies. A good exploitation of heterosis in *G. gynandra* requires the identification of heterotic patterns and groups in the species. To this end, the observed genetic differentiation between accessions based on geographical origin is key, and further investigation to assess the cross-compatibility between them, as well as within each region, is essential to avoid possible incompatibility between accessions. Additionally, identifying familiar testers will help expedite the exploitation of heterosis in the species.

Both additive and nonadditive gene action are controlling minerals, with a predominance of nonadditive genes in *G. gynandra*. More importantly, we observed a positive association between $F_1$ performance and SCA, the sum of both parents' GCA effects and $F_1$ performance, showing that the prediction of $F_1$ performance in spider plant should be based on models involving both GCA and SCA. Specifically, the correlations between $F_1$ performance and SCA and heterosis were strong. Similarly, a more substantial, significant, and positive correlation between SCA and heterosis (MPH and BPH) was reported for agronomic traits in *Brassica oleracea* [85]. The SCA was, therefore, the major driver of hybrid performance and heterosis for leaf mineral content in spider plant and should be considered in selecting parents or populations for improvement. This opens the door to investigating genomic selection methods and machine learning techniques for predicting hybrid performance in the species.

This study revealed that hybridization represents an important breeding strategy for improving the nutritional values of *G. gynandra*. This is attributable to the observed high heterosis, the superiority of specific combining ability over the general combining ability, and the dominance gene action for all leaf mineral element contents investigated (zinc, copper, manganese, calcium, magnesium, sodium, phosphorus, and potassium).

Giving the current global inadequate intakes of calcium (66% of the world population with 5 billion people) and magnesium (31% of the world population with 2.4 billion people) [18] and the high concentration of calcium and magnesium in the leaves of spider plant, the regular consumption of the leaves of spider plant will significantly contribute to combatting calcium and magnesium deficiencies [17]. The consistency of the positive and strong correlation between calcium and magnesium ($r \geq 0.66$, $p < 0.001$) for their concentrations in the leaves of the parents and hybrids, specific combining ability effects, and heterosis, showed that simultaneous selection could be undertaken. Therefore, improving calcium content could result in enhancing magnesium content and could be *the top mineral breeding priority* in *G. gynandra*. A strong correlation between calcium and magnesium concentrations has been previously reported in the species [17,63] and in the leaves of *Brassica napus* L. [97], in shoots of *Brassica oleracea* L. [98]. Possibly, a strong linkage might exist between genes

involved in the accumulation of magnesium and calcium in the leaves of the spider plant, as reported in *Brassica* species [97,98]. For instance, Broadley et al. [98] reported that shoot calcium and magnesium contents in *B. oleracea* are heritable and controlled by pleiotropic loci. Similarly, Alcock et al. [97] also identified loci associated with both calcium and magnesium concentrations in leaves of *B. napus*, which are colocalized in the same regions of the chromosomes. Further studies could investigate the genomic regions controlling the calcium and magnesium accumulation in the leaves of *G. gynandra*. Moreover, this strong association could be explained by the antagonistic interaction between calcium and magnesium in plant cells [99,100]. Consequently, a homeostatic balance between calcium and magnesium ions are important for plant development and growth [99,101]. While calcium play a crucial role for cytosolic signalling, membrane and cell wall integrity [102], magnesium is a key constituent of chlorophyll, essential for photosynthesis, cofactor for enzymes, protein synthesis and energy metabolism [103]. Though calcium and magnesium have different biochemical and physiological roles in plants, homeostasis of calcium and magnesium ions appeared to be closely linked and at least partly regulated by common signaling networks [99]. Specifically, $Ca^{2+}$ signaling participate in regulating the dynamic homeostasis of $Mg^{2+}$ [99]. Therefore, it was suggested that high external magnesium ions would lead to a transient rise in cytosolic calcium ions in plant cells. In case of excess magnesium, the $Ca^{2+}$ signal is detected by two tonoplast-localized calcineurin B-like (CBL) proteins (CBL2 and CBL3), which concomitantly activate four of CBL-interacting protein kinases (CIPKs) (CIPK3, 9, 23 and 26). This CBL–CIPK complex, therefore, recruits $Mg^{2+}$ and regulate downstream specific transporters in the tonoplast for efficient sequestration of $Mg^{2+}$ in vacuoles, thereby ensuring a non-toxic level of $Mg^{2+}$ in the cell as a protective mechanism [104]. This opens rooms for the use of *G. gynandra* to uncover the physiological and regulatory mechanism of calcium and magnesium hemostasis in plants. Furthermore, improving calcium and magnesium content might lead to increased sodium and potassium content, as positive associations were observed between magnesium and sodium for heterosis, SCA effects, and leaf content in hybrids, and between sodium and potassium for SCA effects and hybrids' leaf content.

Breeding for improved iron content in spider plant leaves could be the *second top breeding* priority. Iron is an essential micronutrient and represents the most deficient as its inadequate intake affects about 4.9 billion people, 65% of the world population [18] specifically children and expectant women. Improving the iron content in the leaves of the spider plant could lead to an increase in zinc content (another essential and deficient nutrient affecting 3.5 billion people, or 46% of the world's population) through the indirect effect of phosphorus. Phosphorus and iron were positively correlated in the content of leaves from the parents and hybrids, as well as in specific combining ability effects and heterosis. In addition, phosphorus had a positive association with zinc for their content in the leaves of the hybrids as well as for specific combining ability effects. A positive correlation was previously reported between iron and zinc content in the species, with zinc having a positive direct effect on iron content [17]. Breeding that takes into account the SCA effect of iron could improve the zinc, copper, and phosphorus content, as positive associations were observed between their SCA effects.

As this study pointed out that hybrid cultivars will help improve the species productivity, the establishment of hybrid breeding will be beneficial. However, this might face some challenges, such as the cost-effectiveness of the breeding program and the willingness of farmers to buy hybrid seeds as hybrids seeds are often expensive compared with pure lines or open pollinated cultivars. Given the competitive nutritional advantages of *G. gynandra* over some world leading *Brassica* vegetables [2], a proper advocacy and rising awareness could boost the species demand across Africa and the globe. The success of the development of hybrid cultivars for spider plant will require (i) an investigation of male sterility to ease the hybrid production, (ii) the evaluation of the cost-effectiveness of hybrid cultivars and (iii) the establishment of seed production logistics as vegetable seed system in Africa is still dominated by informal system. Despite these challenges, experience from Brassicaceae species, a sister family to Cleomaceae family, which *G. gynandra* belongs to could be useful. The implementation of hybrid breeding program should consider a holistic approach that consider all stakeholders along the crop value chains. Therefore, actions could include: (i) the evaluation of best parents and crosses for agronomic traits and adaptability though participatory breeding and tricot citizen science taking into account social factors and networks; (ii) farmers training on good agronomic practices; (iii) the development of seed system to ensure permanent

seed availability; and (iv) the assessment of market acceptability, taste and sensory attributes. At the same time, raising awareness and knowledge sharing of nutritional and health benefits of the species activities should be implemented to stimulate demand for market creation.

## Conclusion

The present study has generated critical and novel insights into the genetic mechanisms governing the inheritance of mineral content in *G. gynandra*. We observed significant variation in mineral content among parents and hybrids. Genetic variance components analysis revealed significant effects of both general and specific combining ability, indicating the action of both additive and nonadditive genes, with the predominance of nonadditive gene action in the inheritance of mineral content in the species. The degrees of dominance observed varied depending on the trait, ranging from dominance to overdominance. Our results also revealed the presence of both negative and positive mid- and best-parent heterosis for leaf mineral content in the species. High broad-sense and low to moderate narrow-sense heritability estimates were observed for all minerals. Significant genetic gain was obtained for all mineral contents at a selection pressure of 5%. The best crosses resulted from different parental combinations, ranging from good to poor combiners, suggesting that selection should be based on both general and specific combining ability effects. Heterosis breeding and reciprocal recurrent selection are ideal breeding strategies for developing mineral-dense cultivars that enhance nutrition. Several cultivars can be developed, including hybrids, open-pollinated varieties, and synthetic varieties. We therefore suggest using *G. gynandra* as a model crop for investigating the mechanisms underlying heterosis in plants. Further research on heterotic groups and patterns, as well as tester identification, is needed to fully exploit heterosis in the species. Overall, parents with good combining ability (P05, P09, P10, P11, P12, P15, P24, P25, P26) and crosses expressing promising hybrid vigor (P10×P13, P01×P21, P09×P19, P10×P14) were identified and represent resources for breeding and research purposes.

## Supporting information

**S1 Table. Representation of the North Carolina Design II implemented to generate the 118 F1 hybrids used in the present study.**
(DOCX)

**S2 Table. Phenotypic values of leaf minerals content of 118 experimental hybrids and their 26 parents of *Gynandropsis gynandra* evaluated in 2019 and 2020.**
(XLSX)

**S1 Fig. Variations in temperature, relative humidity and solar radiation under the greenhouse in 2019 and 2020.**
(A) Temperature in 2019. (B) Temperature in 2020. (C) Relative humidity in 2019. (D) Relative humidity in 2020. (E) Solar radiation in 2019. (F) Solar radiation in 2020.
(TIF)

**S2 Fig. Estimates of specific combining ability (SCA) effects of calcium concentration for 118 experimental hybrids of *G. gynandra* evaluated in 2019 and 2020.** ***, **, * refer to estimate of specific combining ability effect significantly different from zero at $p < 0.001$, 0.01 and 0.05, respectively.
(TIF)

**S3 Fig. Estimates of specific combining ability (SCA) effects of magnesium concentration for 118 experimental hybrids of *G. gynandra* evaluated in 2019 and 2020.** ***, **, * refer to estimate of specific combining ability effect significantly different from zero at $p < 0.001$, 0.01 and 0.05, respectively.
(TIF)

**S4 Fig. Estimates of specific combining ability (SCA) effects of phosphorus concentration for 118 experimental hybrids of *G. gynandra* evaluated in 2019 and 2020.** \*\*\*, \*\*, \* refer to estimate of specific combining ability effect significantly different from zero at p < 0.001, 0.01 and 0.05, respectively.
(TIF)

**S5 Fig. Estimates of specific combining ability (SCA) effects of potassium concentration for 118 experimental hybrids of *G. gynandra* evaluated in 2019 and 2020.** \*\*\*, \*\*, \* refer to estimate of specific combining ability effect significantly different from zero at p < 0.001, 0.01 and 0.05, respectively.
(TIF)

**S6 Fig. Estimates of specific combining ability (SCA) effects of sodium concentration for 118 experimental hybrids of *G. gynandra* evaluated in 2019 and 2020.** \*\*\*, \*\*, \* refer to estimate of specific combining ability effect significantly different from zero at p < 0.001, 0.01 and 0.05, respectively.
(TIF)

**S7 Fig. Estimates of specific combining ability (SCA) effects of iron concentration for 118 experimental hybrids of *G. gynandra* evaluated in 2019 and 2020.** \*\*\*, \*\*, \* refer to estimate of specific combining ability effect significantly different from zero at p < 0.001, 0.01 and 0.05, respectively.
(TIF)

**S8 Fig. Estimates of specific combining ability (SCA) effects of copper concentration for 118 experimental hybrids of *G. gynandra* evaluated in 2019 and 2020.** \*\*\*, \*\*, \* refer to estimate of specific combining ability effect significantly different from zero at p < 0.001, 0.01 and 0.05, respectively.
(TIF)

**S9 Fig. Estimates of specific combining ability (SCA) effects of manganese concentration for 118 experimental hybrids of *G. gynandra* evaluated in 2019 and 2020.** \*\*\*, \*\*, \* refer to estimate of specific combining ability effect significantly different from zero at p < 0.001, 0.01 and 0.05, respectively.
(TIF)

**S10 Fig. Estimates of specific combining ability (SCA) effects of zinc concentration for 118 experimental hybrids of *G. gynandra* evaluated in 2019 and 2020.** \*\*\*, \*\*, \* refer to estimate of specific combining ability effect significantly different from zero at p < 0.001, 0.01 and 0.05, respectively.
(TIF)

**S11 Fig. Estimates of mid parent heterosis of calcium concentration for 118 experimental hybrids of *G. gynandra* evaluated in 2019 and 2020.**
(TIF)

**S12 Fig. Estimates of best parent heterosis of calcium concentration for 118 experimental hybrids of *G. gynandra* evaluated in 2019 and 2020.**
(TIF)

**S13 Fig. Estimates of mid parent heterosis of magnesium concentration for 118 experimental hybrids of *G. gynandra* evaluated in 2019 and 2020.**
(TIF)

**S14 Fig. Estimates of best parent heterosis of magnesium concentration for 118 experimental hybrids of *G. gynandra* evaluated in 2019 and 2020.**
(TIF)

**S15 Fig. Estimates of mid parent heterosis of phosphorus concentration for 118 experimental hybrids of *G. gynandra* evaluated in 2019 and 2020.**
(TIF)

**S16 Fig. Estimates of best parent heterosis of phosphorus concentration for 118 experimental hybrids of *G. gynandra* evaluated in 2019 and 2020.**
(TIF)

**S17 Fig. Estimates of mid parent heterosis of potassium concentration for 118 experimental hybrids of *G. gynandra* evaluated in 2019 and 2020.**
(TIF)

**S18 Fig. Estimates of best parent heterosis of potassium concentration for 118 experimental hybrids of *G. gynandra* evaluated in 2019 and 2020.**
(TIF)

**S19 Fig. Estimates of mid parent heterosis of sodium concentration for 118 experimental hybrids of *G. gynandra* evaluated in 2019 and 2020.**
(TIF)

**S20 Fig. Estimates of best parent heterosis of sodium concentration for 118 experimental hybrids of *G. gynandra* evaluated in 2019 and 2020.**
(TIF)

**S21 Fig. Estimates of mid parent heterosis of iron concentration for 118 experimental hybrids of *G. gynandra* evaluated in 2019 and 2020.**
(TIF)

**S22 Fig. Estimates of best parent heterosis of iron concentration for 118 experimental hybrids of *G. gynandra* evaluated in 2019 and 2020.**
(TIF)

**S23 Fig. Estimates of mid parent heterosis of copper concentration for 118 experimental hybrids of *G. gynandra* evaluated in 2019 and 2020.**
(TIF)

**S24 Fig. Estimates of best parent heterosis of copper concentration for 118 experimental hybrids of *G. gynandra* evaluated in 2019 and 2020.**
(TIF)

**S25 Fig. Estimates of mid parent heterosis of manganese concentration for 118 experimental hybrids of *G. gynandra* evaluated in 2019 and 2020.**
(TIF)

**S26 Fig. Estimates of best parent heterosis of manganese concentration for 118 experimental hybrids of *G. gynandra* evaluated in 2019 and 2020.**
(TIF)

**S27 Fig. Estimates of mid parent heterosis of zinc concentration for 118 experimental hybrids of *G. gynandra* evaluated in 2019 and 2020.**
(TIF)

**S28 Fig. Estimates of best parent heterosis of zinc concentration for 118 experimental hybrids of *G. gynandra* evaluated in 2019 and 2020.**
(TIF)

## Acknowledgments

The authors would like to thank Mr Johannes Sibusiso Buthelezi, Senior Laboratory Technician of the Laboratory of Soil Science of the School of Agricultural, Earth and Environmental Sciences, University of KwaZulu-Natal, for his guidance during the laboratory analysis. We are also grateful to Mr Matt Erasmus, Senior Field Technician, the School of Agricultural, Earth and Environmental Sciences, University of KwaZulu-Natal, for providing all the inputs during the greenhouse experiment.

## Author contributions

**Conceptualization:** Aristide Carlos Houdegbe, Enoch G. Achigan-Dako, E. O. Dêêdi Sogbohossou, Alfred O. Odindo, Julia Sibiya.

**Data curation:** Aristide Carlos Houdegbe.

**Formal analysis:** Aristide Carlos Houdegbe.

**Funding acquisition:** Aristide Carlos Houdegbe.

**Investigation:** Aristide Carlos Houdegbe.

**Methodology:** Aristide Carlos Houdegbe, Enoch G. Achigan-Dako, E. O. Dêêdi Sogbohossou, Alfred O. Odindo, Julia Sibiya.

**Project administration:** Aristide Carlos Houdegbe, Enoch G. Achigan-Dako, Julia Sibiya.

**Resources:** Aristide Carlos Houdegbe, Enoch G. Achigan-Dako, E. O. Dêêdi Sogbohossou, M. Eric Schranz, Julia Sibiya.

**Software:** Aristide Carlos Houdegbe.

**Supervision:** Enoch G. Achigan-Dako, Alfred O. Odindo, Julia Sibiya.

**Validation:** Aristide Carlos Houdegbe, Enoch G. Achigan-Dako, E. O. Dêêdi Sogbohossou, Alfred O. Odindo, M. Eric Schranz, Julia Sibiya.

**Visualization:** Aristide Carlos Houdegbe, Enoch G. Achigan-Dako, E. O. Dêêdi Sogbohossou, Alfred O. Odindo, M. Eric Schranz, Julia Sibiya.

**Writing – original draft:** Aristide Carlos Houdegbe.

**Writing – review & editing:** Aristide Carlos Houdegbe, Enoch G. Achigan-Dako, E. O. Dêêdi Sogbohossou, Alfred O. Odindo, M. Eric Schranz, Julia Sibiya.

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
