## [Decision Letter · Decision Letter 0]

27 May 2025

PONE-D-25-18553Combining ability and heterosis analysis for mineral content in the leafy vegetable Gynandropsis gynandra (L.) Briq.PLOS ONE

Dear Dr. Houdegbe,

Thank you for submitting your manuscript to PLOS ONE. After careful consideration, we feel that it has merit but does not fully meet PLOS ONE’s publication criteria as it currently stands. Therefore, we invite you to submit a revised version of the manuscript that addresses the points raised during the review process.

We look forward to receiving your revised manuscript.

Kind regards,

Mehdi Rahimi, Ph.D.

Academic Editor

PLOS ONE

Reviewers' comments:

Reviewer's Responses to Questions

**Comments to the Author**

1. Is the manuscript technically sound, and do the data support the conclusions?

Reviewer #1: Yes

Reviewer #2: Yes

2. Has the statistical analysis been performed appropriately and rigorously? 

Reviewer #1: Yes

Reviewer #2: Yes

3. Have the authors made all data underlying the findings in their manuscript fully available?

Reviewer #1: No

Reviewer #2: Yes

4. Is the manuscript presented in an intelligible fashion and written in standard English?

Reviewer #1: Yes

Reviewer #2: Yes

5. Review Comments to the Author

Reviewer #1: The manuscript titled “Combining ability and heterosis analysis for mineral content in the leafy vegetable Gynandropsis gynandra” presents a comprehensive and technically robust study on the genetic analysis of mineral content using North Carolina Design II. The experimental design is appropriate, the dataset is extensive, and the statistical analyses have been conducted with rigor using REML-based mixed models in ASReml-R. The manuscript addresses an important area in plant breeding and nutrition, particularly for underutilized crops with high micronutrient value.

The findings are well-supported by the presented data. The authors successfully demonstrate the prevalence of non-additive gene action in the inheritance of mineral content and the significant potential of heterosis in improving these traits. The study provides practical insights for hybrid breeding strategies in G. gynandra.The manuscript is of publishable quality after minor revision. I would recommend that you accept the correction of typographical and referencing issues noted in the attached file, and confirm that all underlying data are publicly available in compliance with PLOS ONE policy. Please ensure that the revised version clearly addresses these points.

Reviewer #2: I have completed our review of your manuscript "Combining ability and heterosis analysis for mineral content in the leafy vegetable Gynandropsis gynandra (L.) Briq." While the research addresses an important topic in orphan crop improvement, several major revisions are needed to strengthen the manuscript's impact and scientific rigor.

Introduction

• Add specific data on impact potential, compare to other biofortification efforts, and better justify why this crop deserves genetic research investment compared to established biofortification programs.

• Define GCA/SCA and heterosis concepts clearly for broader readership, discuss breeding implications of the outcrossing nature, and connect to related Brassicaceae breeding work to provide context.

• Replace vague objectives with measurable ones (e.g., "quantify heterosis magnitude" vs "determine gene action"), add missing hypotheses about correlations and stability, and connect each objective to practical breeding decisions.

• The current single-sentence gap statement is too brief and narrow, only mentioning that "no information exists for G. gynandra." This fails to justify why this research matters beyond academic curiosity and should be rewritten to demonstrate clear translational impact potential.

Materials and Methods

• Justify experimental design choices: Why were specific lines assigned as females vs. males? This affects interpretation of GCA estimates and should be explicitly explained.

• Address incomplete factorial: Only 118 of 168 possible crosses succeeded. Explain selection criteria and how this affects design balance and statistical power.

• Improve sampling protocols: The "random collection" of leaf samples needs more specific description (which leaves, plant age, time of day, etc.) for reproducibility.

• Enhance environmental documentation: Provide more detailed greenhouse conditions (daily temperature ranges, humidity fluctuations, light levels) as these affect mineral content.

Results

• Since G×Y variance > genotypic variance, present results by year separately rather than just combined, as the current approach may be masking important patterns.

• Address how the substrate change (soil vs. growing medium) affects the trustworthiness of combining ability estimates and breeding recommendations.

• Perhaps the "year" effects are actually substrate effects. Discuss what this means for practical applications and generalizability.

• Many figures are difficult to interpret. Use clearer labeling, better color schemes, and more intuitive layouts to enhance accessibility.

• Don't just report statistical significance. Discuss what effect sizes mean biologically and for each major finding, explain what it means for breeding strategy and variety development.

Discussion

• Can results from controlled greenhouse conditions with artificial substrates be applied to field breeding programs? This limitation needs explicit discussion.

• The strong Ca-Mg correlation deserves deeper mechanistic explanation beyond just mentioning "pleiotropic loci" - discuss potential physiological and molecular mechanisms.

• How would hybrid breeding actually work for this outcrossing species? Discuss male sterility systems, seed production logistics, and economic feasibility.

• How do these results compare to successful biofortification programs (iron beans, zinc wheat, vitamin A crops)? What can be learned from those experiences?

• Address unique constraints of breeding underutilized species vs. major crops, including market development, value chains, and resource limitations.

6. PLOS authors have the option to publish the peer review history of their article (what does this mean? ). If published, this will include your full peer review and any attached files.

**Do you want your identity to be public for this peer review?** For information about this choice, including consent withdrawal, please see our Privacy Policy .

Reviewer #1: **Yes: ** Ercan CEYHAN

Reviewer #2: No

---

## [Author Response · Author response to Decision Letter 1]

6 Aug 2025

Response to Editor and reviewers' comments (Please also kindly refer to the file Response to Reviewers)

We thank the editor and the reviewers for their valuable and helpful comments on the manuscript. We provided below a point-by-point response to their comments.

Editor’s comments

Comment #1. Please ensure that your manuscript meets PLOS ONE's style requirements, including those for file naming. The PLOS ONE style templates can be found at https://journals.plos.org/plosone/s/file?id=wjVg/PLOSOne_formatting_sample_main_body.pdf and https://journals.plos.org/plosone/s/file?id=ba62/PLOSOne_formatting_sample_title_authors_affiliations.pdf

Answer #1. We thank the editors and reviewers for their valuable comments. PLOS ONE’s style requirements were checked, and the revised manuscript was corrected accordingly.

Comment #2. We note that the grant information you provided in the ‘Funding Information’ and ‘Financial Disclosure’ sections do not match. When you resubmit, please ensure that you provide the correct grant numbers for the awards you received for your study in the ‘Funding Information’ section.

Answer #2. The information in the ‘Funding Information’ and ‘Financial Disclosure’ were crosschecked. The grant number for the award received for this study provided in the ‘Funding Information’ section was corrected to match the information in the ‘Financial Disclosure’.

Reviewer #1

Overall comment:

Point#1. The manuscript titled “Combining ability and heterosis analysis for mineral content in the leafy vegetable Gynandropsis gynandra” presents a comprehensive and technically robust study on the genetic analysis of mineral content using North Carolina Design II. The experimental design is appropriate, the dataset is extensive, and the statistical analyses have been conducted with rigor using REML-based mixed models in ASReml-R. The manuscript addresses an important area in plant breeding and nutrition, particularly for underutilized crops with high micronutrient value.

The findings are well-supported by the presented data. The authors successfully demonstrate the prevalence of non-additive gene action in the inheritance of mineral content and the significant potential of heterosis in improving these traits. The study provides practical insights for hybrid breeding strategies in G. gynandra. The manuscript is of publishable quality after minor revision. I would recommend that you accept the correction of typographical and referencing issues noted in the attached file, and confirm that all underlying data are publicly available in compliance with PLOS ONE policy. Please ensure that the revised version clearly addresses these points.

Answer#1. We thank the reviewer for his valuable and helpful comments on the manuscript. All the correction of typographical and referencing issues provided were accepted (See the revised manuscript).

We confirm that all underlying data are publicly available in compliance with PLOS ONE policy as all relevant data are within the manuscript and its supporting information files. The present study's dataset was uploaded as supporting information files (S2 Table). This was clearly mentioned in the Data Availability Statement as follows: “Data Availability: All relevant data are within the manuscript and its Supporting Information files.”

Reviewer #2

Overall comment: I have completed our review of your manuscript "Combining ability and heterosis analysis for mineral content in the leafy vegetable Gynandropsis gynandra (L.) Briq." While the research addresses an important topic in orphan crop improvement, several major revisions are needed to strengthen the manuscript's impact and scientific rigor.

Point#2.1- Introduction: Add specific data on impact potential, compare to other biofortification efforts, and better justify why this crop deserves genetic research investment compared to established biofortification programs.

Answer#2.1: We thank the reviewer for this suggestion. Data on impact potential, compararison to other biofortification efforts, and justification of why this crop deserves genetic research investment compared to established biofortification programs were included in the revised manuscript with track changes (see lines 92-104) as follows:

“More importantly, G. gynandra leaves have 4.7- and 3.2-fold of vitamin C, 4- and 2-fold iron, 1.2- and 2.1-fold zinc, 2.7- and 10.4-fold calcium, 3.3- and 5.5-fold phosphorus, 1.4- and 1.8-fold potassium, 2.59 and 1.37-total phenolics, and 5.70- and 2.46-fold flavonoids concentrations higher than two world leading commercial and consumed vegetables namely cabbage (Brassica oleracea var. capitata cv. Drumhead) and Swiss chard (Beta vulgaris L. cv. Fordhook Giant), respectively [1]. Breeding efforts in G. gynandra will result in significant impacts on healthier and balanced diets for local communities in Africa and Asia where the species is mostly consumed [2, 3] and high micronutrient deficiencies occurs. Moreover, G. gynandra is a climate resilient crop as it is a C4 plant with the ability to withstand various harsh conditions [3] ; the species was used as a model crop to understand C4 traits, evolution and gene expression in plants [4-6]. The production and the exploitation of the nutritional potential of G. gynandra like many other opportunity crops are limited due to lack of improved varieties resulting from the lack of sustainable breeding program.”

Point#2.2 Introduction: Define GCA/SCA and heterosis concepts clearly for broader readership, discuss breeding implications of the outcrossing nature, and connect to related Brassicaceae breeding work to provide context.

Answer#2.2. We have edited the revised manuscript with track changes (see lines 105-143) as follows:

“Breeding strategies and type of cultivars for a given crop species are guided by the species reproductive biology and mating systems. Both self- and cross-compatibility occurs in G. gynandra with outcrossing being predominant [7-9], offering the possibility for developing both hybrids and inbred cultivars. Because of the predominance of outcrossing, the species might exhibit heterosis and the choice for developing hybrids cultivars sounds as good strategy to exploit the potential heterosis that needs to be properly documented. Heterosis or hybrid vigour refers to the outperformance of the first generation of progenies compared to their parents [10-13] and has been highly researched by breeders for open-pollinated crops but also for self-pollinated crops for higher productivity. Four steps are crucial in the development of a hybrid cultivar: (i) the establishment of populations for selection; (ii) inbred lines development; (iii) inbred lines’ evaluation for combining ability; and (iv) the production of hybrid seed [14]. Following the development of the inbred/advanced lines in G. gynandra [15, 16], their testing for combining ability is the next step towards hybrids creation. Combining ability refers to the ability of a line to combine with another one during hybridization so that desirable genes or traits are transferred to their progenies and encompasses two types, namely the general combining ability (GCA) and the specific combining ability (SCA) [17]. According to Sprague and Tatum [17], a GCA of a line is the average performance of this line in a set of its hybrid combinations and the SCA is the deviation of a specific cross performance from the sum of the average performance or GCA of its parental lines. Combining ability informs on the nature and magnitude of gene action controlling the considered trait. GCA are associated with additive genes effects, while the SCA effects are attributed to nonadditive gene action, including dominance and epistasis gene effects [17]. The GCA helps in the selection of good parents for breeding and the SCA is useful for the selection of the best hybrid combination to better exploit heterosis. The assessment of the combining ability of a set of lines and the inheritance patterns for a target species is done using mating designs such as North Carolina Design II, diallel and line by tester, among others [18-20].

G. gynandra belongs to Cleomaceae family, a sister family of Brassicaceae and could gain more from breeding strategies implemented in Brassica crops. Most developed varieties for Brassica crops nowadays are hybrids and the evaluation of combining ability, heterosis level and gene action has been intensively done for several parents in Brassica oleracea var. capitata [21], Brassica rapa L. [22], and Brassica oleracea var. botrytis L. [23, 24] for nutritional traits such as minerals, antioxidant pigments, and vitamins. Unfortunately, to the best of our knowledge, the heterosis level and the combining ability potential for mineral content in G. gynandra are yet to be investigated to inform breeding program of good parents, heterotic crosses and breeding methods for improved cultivars development.”

Point#2.3. Introduction: Replace vague objectives with measurable ones (e.g., "quantify heterosis magnitude" vs "determine gene action"), add missing hypotheses about correlations and stability, and connect each objective to practical breeding decisions.

Answer#2.3: The objectives were replaced with measurable ones that are connected to practical breeding decisions. The revised objectives were included in the revised manuscript (lines 147-154) as follows:

“Specifically, the study (i) compared the leaf mineral content of 118 experimental F1 G. gynandra hybrids and their parental lines; (ii) quantified heterosis magnitude for leaf mineral content in G. gynandra to select the most suitable breeding method; (iii) evaluated the combining ability effects for leaf mineral content of the parental lines of G. gynandra to identify good parents and best crosses; and (iv) estimated the extent of association between mean performance, heterosis and combining ability for mineral content in G. gynandra to guide prediction of hybrid performance and multiple traits selection.”

The missing hypotheses about correlations and stability were now included in the revised manuscript with track changes (lines 164-166) as follows:

“(v) the leaf mineral elements content in G. gynandra are correlated; and (vi) the correlations between leaf mineral elements content are stable from parents to hybrids.”

Point#2.4. Introduction: The current single-sentence gap statement is too brief and narrow, only mentioning that "no information exists for G. gynandra." This fails to justify why this research matters beyond academic curiosity and should be rewritten to demonstrate clear translational impact potential.

Answer#2.4. The gap statement was revised and could been seen in the introduction section. Please see lines 102-143 of the revised manuscript with track changes.

Point#2.5 Materials and Methods- Justify experimental design choices: Why were specific lines assigned as females vs. males? This affects interpretation of GCA estimates and should be explicitly explained.

Answer#2.5. Previous studies of the reproductive biology revealed that most individual plants of G. gynandra are andromonoecious, producing both staminate and fertile hermaphrodite floral types [7, 8]. The proportion of each flower type on individual plant varied from one population to another, ranging from 48 to 70% for staminate flowers and 30 to 52% for fertile hermaphrodite flowers [7, 8]. Therefore, parental lines with abundant fertile hermaphrodite flowers were considered as females and those with abundant staminate flowers were considered as males. This information is now included in the revised manuscript with track changes (lines 173-180) as follows:

“The lines were separated into two groups; 12 and 14 lines used as females and males, respectively, based on the male/staminate flowers/pollen and female/hermaphrodite flowers production ability (Table 1) as most individual plants in the species were reported to be andromonoecious with differential flowers productivity [7, 8]. Males were crossed with females in a North Carolina design II during two summer seasons (season 1, from October 2018 to February 2019 and season 2, from October 2019 to March 2020) in a greenhouse at the Controlled Environment Facility (29°46′ S, 30°58′ E) of the University of KwaZulu-Natal, Pietermaritzburg Campus, South Africa.”

Point#2.6- Materials and Methods: Address incomplete factorial: Only 118 of 168 possible crosses succeeded. Explain selection criteria and how this affects design balance and statistical power.

Answer#2.6: The crossings between the 12 females and 14 lines males were expected to generate 168 single crosses, but only 118 crosses had sufficient amount of seeds for evaluation. Therefore, the only criteria used was the hybrid seed quantity for planting. This resulted in an incomplete factorial mating design. Luckily, Keuls and Garretsen [25] and Garretsen and Keuls [26] provided a comprehensive framework for the analysis of genetic variation in complete and incomplete diallels and North Carolina II designs. However, their analysis was based on analysis of variance. Here we employed linear mixed-effect model (LMM), a modification of the linear model of ANOVA, due to its advantages over the ANOVA models [27]. Firstly, LMM allows using both fixed and random effects, while ANOVA is employed for only fixed effects. Secondly, a linear mixed effects model provides better results as it handle well missing data, if any. Moreover, LMM provides information not only on the significant effect of phenotypic variation but also on the replication/block. All the factors were set as random, the likelihood ratio test [28] was used to test their significance. This approach has been used in several studies involving incomplete factorial mating design for various crop species, including tomato [29], maize [30], wheat [31-35], sorghum [36].

Point#2.7- Materials and Methods: Improve sampling protocols: The "random collection" of leaf samples needs more specific description (which leaves, plant age, time of day, etc.) for reproducibility.

Answer#2.7: Leaf samples were collected from plants four weeks after transplanting, meaning eight weeks old plants from sowing date. Tenders and edible leaves were randomly picked from plants in the morning between 08:00 am and 11:00 am. This information was included in the revised manuscript as follows (lines 227-229 of the revised manuscript):

“A sample of 20 g of tenders and edible fresh leaves per genotype was randomly collected from all the plants in each replicate in paper bags four weeks after transplanting, meaning eight weeks old plants from sowing date. The leaves were picked from plants between 08:00 and 11:00 am.”

Point#2.8- Materials and Methods: Enhance environmental documentation: Provide more detailed greenhouse conditions (daily temperature ranges, humidity fluctuations, light levels) as these affect mineral content.

Answer#2.8: The greenhouse structure was made of corrugated polycarbonate material. The polycarbonate material that covered the greenhouse reduced the outside solar irradiance by an average 50%. Therefore, the maximum of solar radiation was 356 and 567 W m-2. In 2019, the day temperature ranged between 20 and 35°C and the night temperature ranged between 15 and 25°C. At the same time, the relative humidity was between 75 and 87% at night and between 45 and 70% at day hours. In 2020, while the day temperature ranged between 25 and 38°C and the night temperature ranged between 20 and 25°C. The day relative humidity was between 50 and 80% and between 80 and 90% at night. The hourly day and night variations in temperature, relative humidity fluctuations and solar radiation in the under the greenhouse were presented in S1 Fig.

This information was included in the revised manuscript (lines 218-225) as follows:

“The greenhouse structure was made of corrugated polycarbonate material, which reduced the outside solar irradiance by 50% on average. Therefore, the maximum of solar radiation was 356 and 567 W m-2 in 2019 and 2020, respectively. In 2019, the day temperature varied between 20 and 35°C and the night temperature ranged from 15 to 25°C (S1 Fig). At the same time, the relative humidity was between 75 and 87% at night and 45 and 70% during day hours. In 2020, while t

---

## [Decision Letter · Decision Letter 1]

26 Aug 2025

Combining ability and heterosis analysis for mineral content in the leafy vegetable Gynandropsis gynandra (L.) Briq.

PONE-D-25-18553R1

Dear Dr. Houdegbe,

We’re pleased to inform you that your manuscript has been judged scientifically suitable for publication and will be formally accepted for publication once it meets all outstanding technical requirements.

Kind regards,

Mehdi Rahimi, Ph.D.

Academic Editor

PLOS ONE

Additional Editor Comments (optional):

Reviewers' comments:

Reviewer's Responses to Questions

**Comments to the Author**

1. If the authors have adequately addressed your comments raised in a previous round of review and you feel that this manuscript is now acceptable for publication, you may indicate that here to bypass the “Comments to the Author” section, enter your conflict of interest statement in the “Confidential to Editor” section, and submit your "Accept" recommendation.

Reviewer #1: All comments have been addressed

Reviewer #2: All comments have been addressed

2. Is the manuscript technically sound, and do the data support the conclusions?

Reviewer #1: Yes

Reviewer #2: Yes

3. Has the statistical analysis been performed appropriately and rigorously? 

Reviewer #1: Yes

Reviewer #2: Yes

4. Have the authors made all data underlying the findings in their manuscript fully available?

Reviewer #1: Yes

Reviewer #2: Yes

5. Is the manuscript presented in an intelligible fashion and written in standard English?

Reviewer #1: Yes

Reviewer #2: Yes

6. Review Comments to the Author

Reviewer #1: The authors have carefully addressed the previously recommended revisions. The manuscript has been improved accordingly, and the necessary corrections have been incorporated as suggested.

Reviewer #2: This revision demonstrates substantial improvement in scientific rigor, clarity, and practical relevance. The authors have transformed what was primarily an academic exercise into a manuscript with clear translational impact. The mechanistic insights, breeding recommendations, and comparative context significantly strengthen the contribution.

7. PLOS authors have the option to publish the peer review history of their article (what does this mean? ). If published, this will include your full peer review and any attached files.

**Do you want your identity to be public for this peer review?** For information about this choice, including consent withdrawal, please see our Privacy Policy .

Reviewer #1: **Yes: ** Ercan CEYHAN

Reviewer #2: No

---

## [Editor Report · Acceptance letter]

PONE-D-25-18553R1

PLOS ONE

Dear Dr. Houdegbe,

I'm pleased to inform you that your manuscript has been deemed suitable for publication in PLOS ONE. Congratulations! Your manuscript is now being handed over to our production team.

Kind regards,

on behalf of

Associate Prof. Mehdi Rahimi

Academic Editor

PLOS ONE